# Decoding Layer by Layer: Uncovering Hierarchical Reasoning in Language Models

## Abstract

Decoder-only language models, such as GPT and LLaMA, generally decode on the last layer. Motivated by humans' hierarchical reasoning capability, we propose that a hierarchical decoder architecture could be built with different layers decoding texts in a streaming manner. Due to limited time and computational resources, we choose to adapt a pretrained language model into this form of hierarchical decoder. Language heads of the last layer are copied to different selected intermediate layers, and fine-tuned with different task inputs. By thorough experiments, we validate that these selective intermediate layers could be adapted to speak meaningful and reasonable contents, and this paradigm of hierarchical decoder can obtain state-of-the-art performances on multiple tasks such as hierarchical text classification, classification-guided generation, and hierarchical text generation. HdLM outperforms all baselines on WoS, DBpedia, ESConv, EmpatheticDialogues, AQuA, CommonSenseQA, and several cognitive tests. We also provide a thorough theoretical analysis to validate the convergence and computational savings of our methodology. This study suggests the possibility of a generalized hierarchical reasoner, pretraining from scratch. Our code and model can be found on https://anonymous.4open.science/r/HdLM-2025.

## 1 Introduction

Modern Large language models (LLM), such as GPT (Team, 2024) and Llama (AI@Meta, 2024), have made impressive generalizability and scalability on different types of natural language tasks (Naveed et al., 2024). Among these tasks, reasoning is often challenged, which requires the model to plan the immediate steps and explicitly bootstrap the long-term rewards. To enhance such capabilities, chain-of-thought (CoT) (Jason Wei, 2022) and corresponding finetuning or test-time scaling methods (Eric Zelikman, 2022a;b; Hao et al., 2024) are proposed, with the thought or rationale formulated either on the token-space or the latent-space. However, these 'reasoning LLMs' adhere to the decoder-only architecture, therefore decode the rationale content at the very last layer, prior to the formal response, resulting in significant overhead. Furthermore, their reasoning capabilities are primarily data-driven, constrained by the scaling of annotated data, without an explicit, proactive rationale generation (Yue et al., 2025).

On the contrary, mankind is naturally empowered with hierarchical reasoning capabilities (Murray et al., 2014; Huntenburg et al., 2018; Zeraati et al., 2023), typically in two aspects:

(i) **Multi-timescale of conceptual abstractions**: coarse-grained, strategic decisions (*i.e.*, slow thinking) guide fine-grained, detailed actions (*i.e.*, fast thinking). For example, in emotional support dialogues, the model first determines a high-level strategy (e.g., asking, concluding, or introducing a new topic), then generates granular responses aligned with that strategy (e.g., "How about your feelings?" following the asking strategy).

(ii) **Sequential determination**: upstream actions followed by downstream actions; sometimes the availabilities of downstream options are constrained by the executed upstream action; *e.g.*, in CoT reasoning, the model processes intermediate steps sequentially before arriving at the final answer.

Motivated by both mechanisms, models with hierarchical reasoning capabilities have recently been explored, either implementing on (i) (Barrault et al., 2024; Wang et al., 2025) or (ii) (Yang et al., 2024; Cai et al., 2024). Different from HRM (Cai et al., 2024) which is built on recurrent architectures,

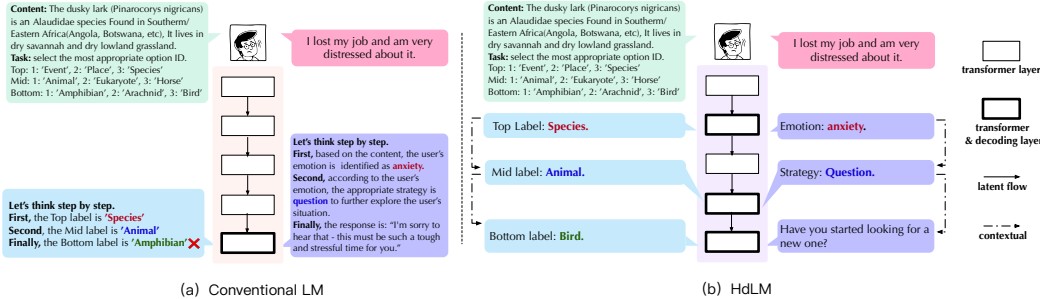

Figure 1: Paradigm of HdLM compared to a conventional decoder-only language model. We exhibit two typical cases: i) example of multiscale concept abstraction, which first recognizes emotion and strategy, then provides the detailed response (in pink); ii) example of sequential determination, with hierarchical label classification (in green). In contrast to the last layer decoding, HdLM employs different layers to decode different levels of texts.

hierarchical reasoning on LLMs is usually implemented on test-time scaling (Eric Zelikman, 2022b), or layer-wise speculative decoding (Elhoushi et al., 2024). However, such a layer-by-layer paradigm has not been coupled with the training directly, especially when coupling with the strategic thinking of mankind.

In this paper, we propose a new type of decoder called Hierarchical decoding Language model (HdLM), which inherits the strong semantic comprehension capability of LLMs, and combines mechanisms (i) and (ii) in a unified paradigm (Figure 1). Previous studies have revealed that intermediate transformer layers have latent states with strong representative capabilities (Zhao et al., 2024; Skean et al., 2025), and can contribute to the final decoding (Elhoushi et al., 2024). Inspired by these observations, we allow multiple layers of HdLM to decode different plausible contents, which form a hierarchical reasoning structure from upstream to downstream generations. During a forward pass, the latent vectors can be passed from the upstream layer to the downstream layer incrementally, such that the upstream generation has a contextual effect on the downstream generation. To verify its effectiveness, we design different test paradigms, including hierarchical text classification (HTC), classification-guided generation (CgG) and hierarchical text generation (HTG), with HdLM surpassing different baselines across different types of tasks and benchmarks. Comparing with the conventional reasoning LLM, HdLM also enjoys computational savings for both training and inference, which are both verified by theoretical derivation and empirical observations. Main contributions are summarized as follows:

(1) We propose HdLM, which can deal with different types of hierarchical tasks, including hierarchical text classification, classification-guided generation, and hierarchical text generation.

(2) We provide a theoretical analysis on the computational benefit of HdLM, as well as a discussion on its convergence.

(3) We conduct substantial experiments to validate HdLM's performances, on both classification and generation metrics, out-of-domain tests, and parameter visualizations.

(4) We provide in-depth discussions on the trade-off between intermediate and final generations, the intermediate layer selection, and further scalability.

## 2 METHOD

In this section, we first formalize the problem, then propose a dual-layer fine-tuning mechanism, and finally a two-pass inference paradigm.

### 2.1 PROBLEM FORMULATION

Given a language model $\mathcal{M}$, its total number of layers is $K$, and the language head is $\mathcal{H}$. Different from traditional query-response tasks, here we try to define a generalized hierarchical textual task. Given a user query $q$, hierarchical textual tasks require the agent to generate a sequence of responses

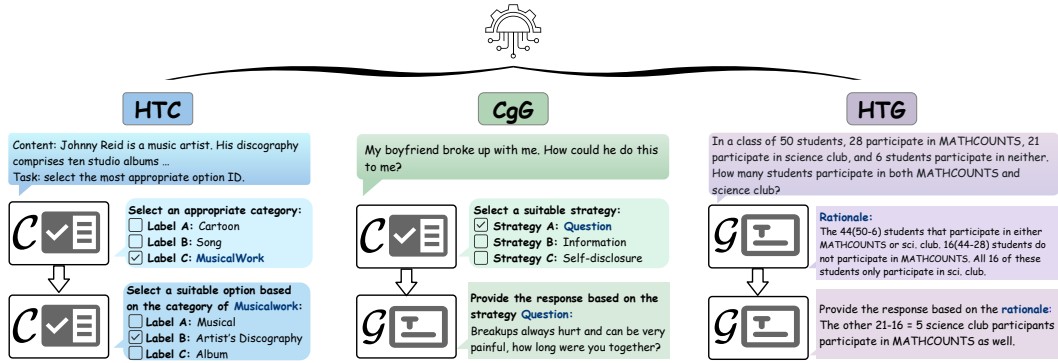

Figure 2: Typical paradigms (hierarchical text classification, classification-guided generation, hierarchical text generation) of hierarchical text tasks. $\mathcal{C}$ denotes classification and $\mathcal{G}$ denotes generation.

$\mathbf{r}_{1:D} := \{r_1, r_2, \cdots, r_D\}$, in which $D$ is the hierarchical depth. The standard format of hierarchical textual data then becomes $(q, \mathbf{r}_{1:D})$. We use $L$ and $L_d$ to denote the lengths of $q$ and $r_d$, respectively.

The above hierarchical textual task can be solved recursively. That is, for each step $i \in \{1, 2, \cdots, D\}$, the current response can be produced grounded by the query and prior responses:

$$\mathcal{T}_i \in \{\mathcal{C}, \mathcal{G}\} : r_i \leftarrow \mathcal{M}(q, \mathbf{r}_{1:i-1}) \tag{1}$$

where $\mathcal{T}$ denotes a uni-step subtask. Here we further argue $\mathcal{T}$ can be generally classified into two categories: the classification task $\mathcal{C}$ and the generation task $\mathcal{G}$. As indicated in Figure 2, the following different paradigms can be summarized from hierarchical textual tasks:

**Hierarchical text classification (HTC):** classifications are asked from the coarse-grained label to the fine-grained label. The lower-level label candidates are usually constrained by the choice of the higher-level label. We denote this paradigm as $\mathcal{C} \rightarrow \mathcal{C}$.

**Classification-guided generation (CgG):** a classification result is first required, then the final response is generated based on that classified label. This paradigm can be denoted as $\mathcal{C} \rightarrow \mathcal{G}$.

**Hierarchical text generation (HTG):** the final response is posterior inferred by intermediate generations, such as the chain of thought, reasoning, or thoughts. This paradigm is denoted as $\mathcal{G} \rightarrow \mathcal{G}$.

**Other hierarchical tasks:** There might be even more complicated hierarchical tasks. For example, the generative classification ($\mathcal{G} \rightarrow \mathcal{C}$) and 'think, classify and act' ($\mathcal{G} \rightarrow \mathcal{C} \rightarrow \mathcal{G}$). We leave these types of tasks for future work.

While there are evident methods to solve hierarchical textual tasks, either by computation time scaling (such as CoT or reasoning LLM), or the multi-hop inference, in this paper, we propose a single-model framework that has less computational overhead.

## 2.2 ARCHITECTURE

In this section, we propose an adjusted architecture based on the decoder-only transformer, which can also deal with hierarchical textual tasks by post-hoc adaptations. We first make a reasonable assumption that the depth of the hierarchical task is smaller, *i.e.*, $D < K$. Then we select $D - 1$ intermediate layers with their indices satisfying:

$$k \in \{k_1, \cdots, k_{D-1}, K\}, 0 < k \leq K \tag{2}$$

These layers, along with the final layer, are used to decode $D$ responses. To achieve this objective, we replicate the language heads in the $K$-th layer to the $D - 1$ intermediate layers, with the parameter randomly initialized.

$$\mathcal{H}_d \leftarrow \mathcal{H}_K, d = 1, 2, \cdots, D - 1 \tag{3}$$

Figure 3: The pipeline of HdLM, with exampled $D = 3$ and $K = 5$. (a): A depth=3 hierarchical task on a conventional LM. (b): Replicate the language heads of the $K$-th layer to layer $k_1, k_2$. (c): Training HdLM with the fine-tuning loss of $r_d$ on the $k_d$-th layer. (d): Inference HdLM on the $k_d$-th layer sequentially with decoded $\hat{r}_d$ and corresponding length $\hat{L}_d$.

To make the derivation clearer, here we propose some notations. Given a language model $\mathcal{M}$, $\mathcal{M}_k$ represents its $k$-th layer. The forward computation of latent vector from the $k_1$ to $k_2$-th layer is denoted by $\mathcal{M}_{k_1:k_2}$, while $\mathcal{M}_{k_1:k_2}^{\xrightarrow{L_1}}$ denotes the forward computation plus a textual generation of $L_1$ at the $k_2$-th layer.

The final layer still holds the standard supervised Fine-Tuning (FT) loss:

$$\mathcal{L}_D^{\text{FT}} = -\frac{1}{L_D} \sum_{j=1}^{L_D} \log \left[ P(r_D(j)|q, \mathbf{r_{1:D-1}}, r_D(1:j-1)) \right] \tag{4}$$

where $r_D(j)$ denotes the $j$-th token of $r_D$.

## 2.3 TRAINING

The $D - 1$ intermediate layers can not decode reasonably with the newly added heads. To educate them to decode proficient language, post-hoc adaptation is needed.

Given the hierarchical textual sample $(q, r_{0:d})$, the latent vector of each hierarchical level can be calculated recursively by the forward pass of each LM block:

$$e_1 = \mathcal{M}_{0:k_1}(q), \quad e_{d+1} = \mathcal{M}_{k_d:k_{d+1}}(e_d, r_d), \quad \mathcal{L}_d = \mathcal{L}^{\text{FT}}(q, e_{1:d}, r_{1:d}), \quad d = 1, 2, \cdots, D \tag{5}$$

with the finetuning loss $\mathcal{L}_d$ implemented on each decoding layer. The final loss can then be the linear combination of them: $\mathcal{L} = \sum_{d=1}^{D-1} f_d \mathcal{L}_d + \mathcal{L}_D$ with the list of $\{f_1, \cdots, f_{D-1}, 1\}$ as the loss weights. During the training, we implement the causal mask for both prior tokens and subsequent responses. Figure 3 exhibits the detailed masking matrices for all decoding layers.

## 2.4 INFERENCE

We keep a similar forward logic of inference to the training. For the query $q$, we still calculate its logits throughout all attention layers, similar with the conventional language model. For each $k$-th layer, the prior latent vector is employed to decode the $k$-th response, while the new decoded latent is also employed to forward pass the final layer, until the final response is decoded grounded by all previous latents.

$$e_1, \hat{L}_1 = \mathcal{M}_{0:k_1}^{\xrightarrow{L_1}}(q), \quad e_{d+1}, \hat{L}_{d+1} = \mathcal{M}_{k_d:k_{d+1}}^{\xrightarrow{L_d}}(e_d) \tag{6}$$

$$\hat{r}_d = \mathcal{H}_{k_d}(e_d(L + \sum \hat{L}_{<d-1} :)), \quad d = 1, 2, \cdots, D \tag{7}$$

Note the above expression has a similar structure to the state-space model (SSM).

## 3 COMPUTATIONAL COMPLEXITY ANALYSIS

In this section, we compare the computation FLOPs of our HdLM with a standard LLM, and verify that HdLM results in both training and inference reduction. To make the formulation simple and clear, here we discuss the 2-depth hierarchical task, *i.e.*, $D = 2$.

**Parameter definitions.**   The input sequence length $L_i$, the output sequence lengths $L_1$ and $L_2$, the hidden dimension $E$, indexes of two decoding layers $k_1 < K$ and $k_2 = K$, and the expand ratio of FFN dimension is $c$. We use $\mathcal{F}$ to denote the FLOPs.

**Training FLOPs savings.**   We begin with the forward FLOPs of a standard transformer from Equation 20 derived in Appendix B.1: $\mathcal{F}_{\text{train}} \rightarrow 3(8 + 4c)E^2 KL$, where $L$ is the total length of input and output. [1]. The LLM baseline simply treats the input, level-1 output and level-2 output as a flat sequence, with all layers engaged:

$$\mathcal{F}_{\text{train}}^{\text{baseline}} \approx 3fK(L_i + L_1 + L_2) \tag{8}$$

where we use $f$ to represent the fixed coefficient $(8 + 4c)E^2$ for simplicity. On the other hand, the training FLOPs of HdLM can be expressed as

$$\mathcal{F}_{\text{train}}^{\text{HdLM}} \approx 3fk_1(L_i + L_1) + 3f(K - k_1)(L_i + L_1 + L_2) \tag{9}$$

The training FLOPs savings then become

$$\mathcal{F}_{\text{train}}^{\text{baseline}} - \mathcal{F}_{\text{train}}^{\text{HdLM}} \approx 3fk_1 L_2 > 0 \tag{10}$$

**Inference FLOPs savings.**   The inference FLops of LLM baseline can be expressed as

$$\mathcal{F}_{\text{infer}}^{\text{baseline}} = fK \sum_{j=1}^{L_1 + L_2} (L_i + j - 1) \tag{11}$$

from Equation 21 derived in Appendix. While for HdLM, the inference FLOPs becomes

$$\mathcal{F}_{\text{infer}}^{\text{HdLM}} = fk_1 \sum_{j=1}^{L_1} (L_i + j - 1) + f(K - k_1) \sum_{j=1}^{L_2} (L_i + L_1 + j - 1) \tag{12}$$

and it is easy to further prove $\mathcal{F}_{\text{infer}}^{\text{baseline}} - \mathcal{F}_{\text{infer}}^{\text{HdLM}} > 0$ (detailed derivation in Appendix D.1).

## 4 EXPERIMENT

In this section, we consider answering the following research questions: **RQ1:** Can HdLM outperform conventional baselines on different types of hierarchical textual tasks, and how the hierarchical contexts work? **RQ2**: Can HdLM generalized well on out-of-domain hierarchical textual datasets? **RQ3:** How to select the decoding intermediate layers, and how does this choice trade off the generation qualities of different layers? **RQ4:** How well can HdLM scale to different model sizes, dataset sizes, and hierarchical depths?

### 4.1 DATASETS

We first list the datasets we use for different types of hierarchical tasks:

**HTC ($\mathcal{C} \rightarrow \mathcal{C}$).**   We employ the famous WoS (du Toit et al., 2024) (with depth $D = 2$) and DBpedia (Auer et al., 2007) (with depth $D = 3$) as benchmarks, where each depth of task is a textual classification ($\mathcal{C}$). WoS are abstracts of published papers from Web of Science, while DBpedia extracts structured information from Wikipedia.

---

[1]Here we omit the batch size term $B$ since different samples' computation within a batch is parallel.

**CgG** ($\mathcal{C} \rightarrow \mathcal{G}$). We adopt the empathetic conversation scenario, in which each turn of conversation is annotated with the user's emotion and the assistant's response strategy. The model can sequentially classify the $emotion$ ($\mathcal{C}_{emo}$) then the $strategy$ ($\mathcal{C}_{stra}$), and finally generate the $response$ ($\mathcal{G}_{resp}$). We utilize ESConv (Liu et al., 2021) as both training and in-domain (ID) test sets. To further validate the response generation, EmpatheticDialogues (Rashkin et al., 2019) is employed as an out-of-domain (OOD) test set which lacks strategy annotations.

**HTG** ($\mathcal{G} \rightarrow \mathcal{G}$). The model needs first generate a $rationale$ ($\mathcal{C}_{rationale}$) then generate the final $answer$ ($\mathcal{C}_{answer}$) grounded by the rationale. We first employ the reasoning benchmark AQuA (Ling et al., 2017) as the ID test, and CommonSenseQA (Talmor et al., 2019) as the OOD test.

## 4.2 Setting

**Implementation Details.** We conduct a post-hoc adaptation on the basis of Llama3-8B-Instruct (Grattafiori et al., 2024), which has a total $K = 32$ attention layers. During training, we use the AdamW optimizer with decay of 0.01 and the cosine scheduler. The training batch size is 16 and the sequence length is 2048. The experiment is running on LlamaFactory (Zheng et al., 2024) with 32 A100 GPUs, lasting about 16 hours. Other dataset-wise settings are listed in Appendix C.3.

**Classification metrics.** We explore F1-related scores including Micro-F1 (MiF1) and Macro-F1 (MaF1). MiF1 considers the overall precision and recall of all instances, while MaF1 equals the average F1-score of labels. For CgG tasks, we also provide the classification accuracy, and the preference $bias$ as defined by (Kang et al., 2024) based on the Bradley-Terry model (Bradley & Terry, 1952) [2]. For HTC, classification is calculated on the bottom level of labels.

**Generation metrics.** We utilize the famous metrics of BLEU-2 (**B-2**) (Papineni et al., 2002), Rouge-L (**R-L**) (Lin, 2004) and CIDEr (**CDr**) (Vedantam et al., 2015). B-2 first computes the geometric average of the modified $n$-gram precisions, then calculates the brevity penalty and the final score. R-L uses F-measure based on the longest common subsequence to estimate the similarity between two summaries. CDr calculates the cosine similarity from the average of different $n$-grams.

## 4.3 Results

To address **RQ1** and **RQ2**, here we present main results of HTC, CgG and HTG, including both ID and OOD metrics. Appendix D.2 analyzes the losses and Appendix D.8 visualizes the parameters.

**Hierarchical Textual Classification.** Table 1 shows the F1-scores of WoS (depth=2) and DBpedia (depth=3) on their bottom-level labels, compared to previous state-of-the-art HTC baselines. Our HdLM performs the best on both WoS and DBpedia, indicating it has a reasonable hierarchy comprehension and can adapt to different depths. For typical cases of HTC, see Appendix D.5.

**Classification-guided Generation.** In this scenario, we consider baselines including Direct (direct inference), Refine (revise the initial response immediately), Self-Refine (Madaan et al., 2023) (multi-hop inference with refinement on the first-time generation), CoT (Wei et al., 2022) and finite state machine (FSM) (Wang et al., 2024).

Table 2 shows that HdLM also has a reasonable performance on both classification and generation metrics, either the largest or the second-largest. HdLM is also robust to different classification paradigms (with or without emotion) and generalizes well on OOD situations (EmpatheticDialogues). For human evaluations, LLM-as-a-Judge, and typical cases of CgG, see Appendix D.5.

**Hierarchical Textual Generation.** In this experiment, we compare with different reasoning paradigms, including LayerSkip (Elhoushi et al., 2024) and SL-D (Zhu et al., 2024a).

Table 3 indicates that HdLM performs well on these different types of reasoning benchmarks, and generalizes well on OOD situations. For typical cases of HTG, see Appendix D.5.

---

[2]Detailed formula in the Appendix. Smaller $bias$ means better.

Table 1: F1 scores of hierarchical textual classification on WoS and DBPedia. The best results of each model are **bolded** and the second best are underlined.

| | Dataset($\rightarrow$) | WoS ($\mathcal{C} \rightarrow \mathcal{C}$) | | DBPedia ($\mathcal{C} \rightarrow \mathcal{C} \rightarrow \mathcal{C}$) | |
|---|---|---|---|---|---|
| | Method ($\downarrow$) | MiF1 | MaF1 | MiF1 | MaF1 |
| Retrieval-based | HierVerb (Ji et al., 2023) | 80.93 | 73.80 | 96.17 | 93.28 |
| | Retrieval (Chen et al., 2024) | 81.12 | 73.72 | 96.22 | 93.37 |
| | Retrieval-ICL (Chen et al., 2024) | 78.62 | 69.56 | 95.56 | 92.04 |
| Bert-based | BERT | 86.28 | 80.58 | 95.31 | 89.16 |
| | + HiAGM (Zhou et al., 2020) | 86.04 | 80.19 | - | - |
| | + HiMatch (Chen et al., 2021) | 86.70 | 81.06 | - | - |
| | + softprompt (Wang et al., 2022b) | 86.57 | 80.75 | - | - |
| | HGCLR (Wang et al., 2022a) | 87.11 | 81.20 | 95.49 | 89.41 |
| | HPT (Wang et al., 2022b) | 87.16 | 81.93 | 96.13 | 93.34 |
| | SFT | 86.64 | 85.86 | 96.15 | **96.50** |
| | **HdLM** (ours) | **88.40** | **87.54** | **96.73** | 96.37 |

Table 2: Results of classification-guided generation on ESConv (ID) and EmpatheticDialogues (OOD), including classification metrics such as Accuracy (ACC), Macro-F1 (MaF1) and $bias$, and generation metrics such as BLEU-2 (B-2), ROUGE-L (R-L) and CIDEr. Among the task subscripts, 'emo' denotes emotion, 'stra' denotes strategy, and 'resp' denotes response. The best results of each model are **bolded** and the second best are underlined.

| Method | ESConv ($\mathcal{C}_{emo} \rightarrow \mathcal{C}_{stra} \rightarrow \mathcal{G}_{resp}$) | | | | | EmpatheticDialogues ($\mathcal{C}_{stra} \rightarrow \mathcal{G}_{resp}$) | | |
|---|---|---|---|---|---|---|---|---|
| | ACC $\uparrow$ | MaF1 $\uparrow$ | $bias \downarrow$ | B-2 $\uparrow$ | R-L $\uparrow$ | B-2 | R-L | CDr |
| Direct | 11.80 | 10.26 | 1.61 | 3.47 | 10.64 | 3.09 | 9.91 | 1.60 |
| + Refine | 17.08 | 11.07 | 1.27 | 3.10 | 6.13 | 2.56 | 9.12 | 0.42 |
| + Self-Refine | 17.58 | 13.61 | 1.92 | 3.34 | 9.71 | 3.08 | 9.91 | 1.56 |
| + CoT | 15.32 | 10.38 | 1.69 | 3.16 | 10.50 | 2.91 | 9.79 | 1.37 |
| + FSM | 17.37 | 11.15 | **0.81** | 4.12 | 11.83 | 3.33 | 10.80 | 2.96 |
| SFT | 30.60 | 21.29 | 1.28 | 6.97 | 16.59 | 4.10 | 10.88 | 6.84 |
| + CoT | 30.80 | 17.70 | 1.35 | 6.51 | 15.00 | 5.05 | 14.06 | **12.01** |
| + FSM | 28.83 | 18.36 | 1.32 | **7.57** | 17.42 | 4.69 | 14.28 | 10.12 |
| + **HdLM** (ours) | **33.41** | **21.65** | 0.89 | 7.54 | **18.13** | **5.48** | **14.53** | 11.95 |

Table 3: Accuracies (in percentage) on reasoning benchmarks, including AQuA (ID) and Common-SenseQA (OOD). The best results of each model are **bolded** and the second best are underlined.

| Method | HTG ($\mathcal{G}_{rationale} \rightarrow \mathcal{G}_{answer}$) | |
|---|---|---|
| | AQuA (ID) | CommonSenseQA (OOD) |
| 7-shot | 17.7 | 70.9 |
| CoT | 24.8 | 71.4 |
| SFT | 26.9 | 69.6 |
| LoRA | 21.9 | 70.9 |
| LayerSkip (Elhoushi et al., 2024) | **38.5** | 71.3 |
| SL-D (Zhu et al., 2024a) | 38.2 | 67.5 |
| **HdLM** (ours) | 25.2 | **72.9** |

## 4.4 DISCUSSION

**Selection of the intermediate decoding layer.** As studied by Zhao et al. (2024), middle layers encode valuable representations for downstream tasks, and the 32-layer LLaMA-8B model can be partitioned into three segment:

- the region of **shared layers: $1 \leq D \leq 9$.
- the region of **transition layers: $10 \leq D \leq 15$.
- the region of **refinement layers: $16 \leq D \leq 32$.

therefore, the optimal layer selection (for content decoding) should fall within the refinement region ($16 \leq D \leq 32$). Furthermore, Skean et al. (2025) observes that LLM first undergoes a 'compression valley' where entropy decreases until $D = 25$, followed by entropy increase; Zhu et al. (2024a) identifies D = 25 as the intersection point between the 'concept generation' and 'token generation' phases. These findings collectively justify narrowing the optimal D range to around $25 \leq D \leq 32$. This narrowed region allows us to conduct empirical analysis to finally determine the optimal $D$ with much less costs.

To finally address **RQ3**, we conduct a sensitivity study on the choice of intermediate decoding layer, which directly affects the trade-off between intermediate and final response qualities. To make the expression simple, here we constrain the task depth $D = 2$; subsequently, the responses are $(r_1, r_2)$ and we just use $k$ instead of $k_1$ to denote the intermediate decoding layer index.

Figure 4 shows the metric curves on WoS and ESConv, with respect to different choices of $k$. When $k$ is small, the performance is relatively low since the layer-depth of HdLM is not enough for $r_1$ understanding and generation. As $k$ increases, performances also improve. However, when $k$ is close to $K$, they degrade again the layer-depth for $r_D$ starts to be limited. From these results, we choose $k = 25$ for WoS and $k = 28$ for ESConv, which align precisely with the theoretical guidance from (Zhao et al., 2024; Skean et al., 2025; Zhu et al., 2024a). Based on these observations, we also suppose that **the generation subtask may need larger layer depth and higher loss weights to converge, compared to the classification subtask**. Appendix D.7 provides the sensitivity analysis on loss weights.

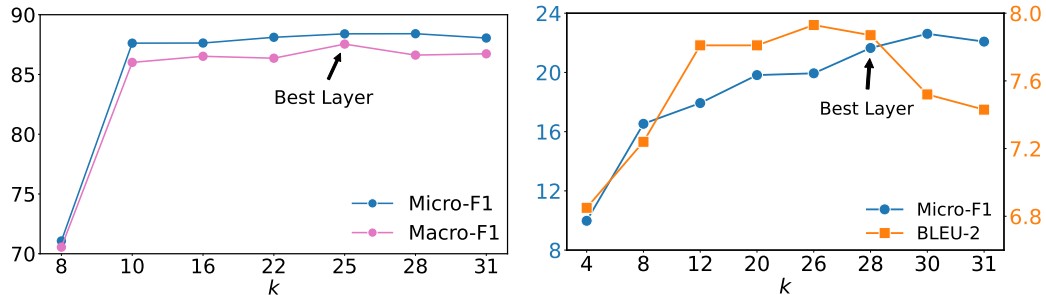

Figure 4: Performance sensitivities on WoS (left) and ESConv (right) with different decoding layer selection ($k$).

**Scalability on model sizes.** As part of **RQ4**, here we explore the model scalability of HdLM. Table 4 shows the WoS results on 1B, 8B and 70B, which reveal that HdLM has a consistent performance advantage compared to SFT. This observation reveals that HdLM can adapt to a variety of model sizes with robust performances.

Table 4: F1 scores of WoS ($\mathcal{C} \rightarrow \mathcal{C}$) experiments based on different sizes of Llama models.

| Size($\rightarrow$) | Llama3-1B-Instruct | | Llama3-8B-Instruct | | Llama3-70B-Instruct | |
|---|---|---|---|---|---|---|
| Method ($\downarrow$) | MiF1 | MaF1 | MiF1 | MaF1 | MiF1 | MaF1 |
| Direct | $\star$ | $\star$ | 29.07 | 31.93 | 10.30 | 8.06 |
| +SFT | 83.61 | 82.97 | 86.64 | 85.86 | 87.65 | 87.26 |
| **HdLM** (ours) | **84.07** | **83.71** | **88.40** | **87.54** | **89.15** | **88.96** |

$\star$: Llama3-1B-Instruct fails to generate reasonable responses in the zero-shot test of WoS.

**Ablation on hierarchical levels.** We further answer **RQ4** on the depth scalability study. Compared to the standard pipeline of ESConv, which has a 3-depth hierarchy $\mathcal{C}_{emo} \rightarrow \mathcal{C}_{stra} \rightarrow \mathcal{G}_{resp}$, we conduct two ablations, *w/o emotion* and *w/o strategy*. Results are compared in Table 5, which indicates that although the 3-depth HdLM generally performs the best, HdLM adapts well to two 2-depth ablations with comparative performances.

Table 5: Ablation results of classification-guided generation on ESConv (ID) and EmpatheticDialogues (OOD), including classification metrics (on strategy) such as Accuracy (ACC), miF1 and $bias$, and generation metrics such as B-2, R-L and CDr.

| Method | ESConv (ID) | | | | | EmpatheticDialogues (OOD) | | |
|---|---|---|---|---|---|---|---|---|
| | ACC ↑ | miF1 ↑ | $bias$ ↓ | B-2 ↑ | R-L ↑ | B-2 ↑ | R-L ↑ | CDr ↑ |
| w/o emotion | 30.34 | 21.14 | **0.43** | 6.83 | 16.35 | 5.32 | 14.43 | 11.92 |
| w/o strategy | N/A | N/A | N/A | 7.37 | 17.3 | 5.47 | **15.60** | 7.74 |
| **HdLM** | **33.41** | **21.65** | 0.89 | **7.54** | **18.13** | **5.48** | 14.53 | **11.95** |

**Computational time.** We also calculate the computational times in the experimental environment. For different $L_2$ (128, 256, 512), the computation reduction of HdLM grows linearly (7.03%, 18.03%, 26.25%), no matter what the input length is. On the other hand, for different $k$ (16, 20, 24), the computation reduction of HdLM also grows linearly (7.43%, 12.09%, 17.85%). These observations verify the theoretical conclusion, revealing the good scalability on the large dataset size. Detailed plots and corresponding error bars are included in Appendix D.6.

## 5 RELATED WORK

**Large language models with latent reasoning pace.** Token-level LLMs are repurposed to reasoning on the latent space to further enhance the thinking capability. Recently, COCONUT (Hao et al., 2024) utilizes the latent state of the LLM to represent the reasoning state, which forms a continuous thought. TRICE (Hoffman et al., 2023) samples and finetunes the chained rationales by Monte-Carlo sampling. LCM (Barrault et al., 2024) studies the sentence-level conceptual embeddings. However, these studies generally reason and decode the latent thoughts on the final layer, which lacks of a structured hierarchical view; while our method constructs a recursive chain of layer latent which is a natural hierarchal thinker.

**Hierarchical Decoding.** There were early attempts at a hierarchical decoding mechanism. Cascade decoder (Liang et al., 2019) employs a cascade branching structure on the biomedical image segmentation tasks. CoHD (Luo et al., 2024) proposes a counting-aware hierarchical decoding framework for image segmentation. Su et.al (Su et al., 2018) introduce a hierarchical decoding NLG model based on different levels of linguistic patterns. HSD (Zhu et al., 2024b) adaptively skips decoding layers in a hierarchical manner. HRM (Cai et al., 2024) implements a two-layer recurrent hierarchical model with approximately 27M parameters, training from scratch. There are more studies on speculative decoding or early layer skipping. For example, LayerSkip (Elhoushi et al., 2024) enables the LLM to decode self-speculatively and inference with early exit; Zhu et al. (2024a) implements the layer skipping with layer contrastive decoding; Medusa (Cai et al., 2024) implements multiple decoding heads in parallel.

On the other hand, our HdLM generates different level of abstractions, in contrast with speculative decoding methods (Elhoushi et al., 2024; Zhu et al., 2024a; Cai et al., 2024); furthermore, HdLM is built on pretrained LLMs, with stronger capabilities and more levels of decoding hierarchies, compared to HRM (Cai et al., 2024).

## 6 CONCLUSION

In this study, we propose a hierarchical decoding language model called HdLM, which can provide both sequential and strategic understanding and generation capabilities. Post-hoc adapted from pretrained language models, HdLM achieves state-of-the-art performance on hierarchical text classification, classification-guided generation, and hierarchical text generation. We also conduct theoretical analysis on its computational efficiency, convergence, and guidance of intermediate layer selection. HdLM sheds some light on a generalized, hierarchical reasoner based on language models.

ETHICS STATEMENT

HdLM provides an automatic framework to allow the LLM to think, plan and reason sequentially, maybe from coarse-grain to fine-grain concepts. Currently, HdLM is a data-driven finetuning framework, which means its output could be safe given that the dataset content is secured. However, there is a possibility that HdLM might generalize to unexpected domains and make harmful plans. Generated thoughts of HdLM should be monitored.

REPRODUCIBILITY STATEMENT

We have made extensive efforts to ensure the reproducibility of our work. The complete implementation of HdLM, including code base, training scripts, and configuration files, has been submitted in the supplementary materials via a GitHub repository link. All datasets used in our experiments are publicly available, and their sources are clearly documented in the appendix.

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

## A  USAGE OF LLM

There is no usage of LLM or other AI tools within the writing process.

## B  PRELIMINARY

### B.1  FLOPS COMPUTATION OF LANGUAGE MODEL

**Parameter definitions.**  We denote key parameters as follows: the batch size $B$, input token length $L$, output token length $L^o$, hidden dimension $E$, the vocabulary size $V$, number of transformer layers $K$, the dimension coefficient $c$ of the FFN intermediate layer. We use $\mathcal{F}$ to represent the computation flops.

**FLOPs of self-attention.**  The computational complexity of the self-attention module is defined as:

$$\mathcal{F}_{\text{train}}^{\text{ATTN}}(L) = 8BLE^2 + 4BL^2E \tag{13}$$

$$\mathcal{F}_{\text{infer}}^{\text{ATTN}}(L,t) = 4BE^2 + 4BE(L+t-1), t = 1, \cdots, L^o \tag{14}$$

where infer is the abbreviation of inference, and $t$ denotes the $t$-th decoded token.

**FLOPs of FFN.**  The feed-forward network's computational complexity is expressed as:

$$\mathcal{F}_{\text{train}}^{\text{FFN}}(L) = 4BcLE^2 \tag{15}$$

$$\mathcal{F}_{\text{infer}}^{\text{FFN}}(L,t) = 4Bc(L+t-1)E^2 \tag{16}$$

**Decoding Layer.**  With the vocabulary size as $V$, the flops of decoding layer is

$$\mathcal{F}^{\text{Decode}}(L) = 2LEV \tag{17}$$

**Forward pass of transformer.** FLOPs of the transformer is the summation of flops of each self-attention layer, FFN layers, plus the decoding layer.

$$
\begin{aligned}
\mathcal{F}_{\text{forward}} &= K(\mathcal{F}_{\text{train}}^{\text{ATTN}}(L) + \mathcal{F}_{\text{train}}^{\text{FFN}}(L)) + \mathcal{F}^{\text{Decode}}(L) \\
&= BK((8 + 4c)LE^2 + 4L^2E) + 2BLEV
\end{aligned}
\tag{18}
$$

In general cases, it is reasonable to assume $E >> L, (4 + 2c)KE >> V$ [3]. Equation 18 then can be reduce to

$$
\mathcal{F}_{\text{forward}} \rightarrow BL(8 + 4c)KE^2
\tag{19}
$$

Equation 19 can be understood from another perspective. $BL$ can be viewed as the total number of input tokens, while the total number of transformer parameters can be approximated to $(4 + 2c)KE^2$, therefore the training flops can also be considered as 2 times number of input tokens times model parameters.

**Training and inference of transformer.** A single step training computation includes a forward pass and two backward propagation. As a result, the training FLOPs is 3 times the forward FLOPs:

$$
\mathcal{F}_{\text{train}} = 3\mathcal{F}_{\text{forward}} \rightarrow 3BL(8 + 4c)KE^2
\tag{20}
$$

while the inference FLOPs needs the accumulation throughout the decoded sequence:

$$
\mathcal{F}_{\text{infer}} = \sum_{j=1}^{L_o} \mathcal{F}_{\text{forward}}(L + j - 1) \rightarrow B(8 + 4c)KE^2 \sum_{j=1}^{L_o}(L + j - 1)
\tag{21}
$$

where $L^o$ is the decoded sequence length.

## B.2 ASSUMPTIONS OF THEOREM 1

Below are the assumptions proposed in (Mușat, 2025) which are prerequisites of Theorem 2.

**Assumption 1.** During self-attention, a position can only attend to another position if they already share a piece of information.

**Assumption 2.** When a position attends to another position, it retrieves all the information contained in the attended position.

## B.3 CONVERGENCE OF DECODER WITH ARBITRARY LAYERS

In this subsection, we propose Corollary 3 to answer the convergence issue, *i.e.*, how can we ensure any implementation decoding layer (with the index of $k_d$), has enough depth to learn the target semantic hierarchy (with depth $d$)? Our derivation is mainly motivated by the Minimum Number of Layers Theorem proposed in (Mușat, 2025),

**Corollary 1.** *Assume Theorem 2 holds in $\mathcal{C}$ or $\mathcal{G}$, then for any $d \leq D$, the first $k_d$ layers of transformer can always learn the top $d$-depth task information.*

Our derivation is mainly motivated by the Minimum Number of Layers Theorem proposed in (Mușat, 2025)

**Theorem 2.** *The last position in the sequence cannot retrieve the embedding vector $x_D$ of the target token with $K$ transformer layers if $K < \log_3(2D)$.*

Due to the page limit, here we leave the detailed assumption of Mușat (2025) and the connection between retrieval and classification in Appendix B.2. Based on its conclusion, here we prove our Corollary 3:

**Corollary 3.** *Assume Theorem 2 holds in $\mathcal{C}$ or $\mathcal{G}$, then for any $d \leq D$, the first $k_d$ layers of transformer can always learn the top $d$-depth task information.*

---

[3] For Llama3 8B, $K = 32, E = 4096, V = 128256$. $K * E$ is similar to $V$ while $4 + 2c$ ¿ 1. For Llama with large sizes, $V$ keeps the same while $K$ and $E$ increase. Therefore, the inequality holds.

*Proof.* Given the strict increasing, non-repeat integer sequence $\{k_d\}$ with $d \in \{1, 2, \cdots, D\}, k \in \{1, 2, \cdots, K\}, D \leq K$, it is obvious to have $k_d \geq d$. Since $3^d > 2d$ holds for $d \geq 1$, then we have $k_d \geq d > \log_3 2d$, which contradicts the situation in Theorem 2. □

Corollary 3 only ensures the theoretical feasibility of our approach, but also provide the guidance of $k_d$ selection.

## C   MORE EXPERIMENTAL CONFIGURATIONS

### C.1   BENCHMARKS

Theory of Mind (ToM) (Premack & Woodruff, 1978) evaluates humans' cognitive ability to attribute mental states, beliefs and desires, especially concurring with others. ToMI (Le et al., 2019) and BigToM (Gandhi et al., 2024) benchmarks are then proposed to test LLMs based on the Sally-Anne false-belief tests. In this scenario, LLM is assigned a specific role and faces a multi-role scenario. Information is provided from different roles' perspectives while LLM should conclude only from the ego-centric perspective.

To validate the ToM capability, we split ToMi and BigToM into training and test sets, and collect the test pass rate of HdLM. As baselines, we compare with direct or COT inference LLMs, standard SFT, and SimTom (Alex Wilf, 2022) which has a two-stage perspective-taking prompt specifically designed for ToM tests.

Vignette-based problem is "a hypothetical situation, to which research participants respond thereby revealing their perceptions, values, social norms or impressions of events.", as indicated by Wikipedia. Binz (Binz & Schulz, 2023) collects a set of 24 Vignette-based questions, covering decision-making, information search, deliberation, causal reasoning, and adversarial confusing abilities.

### C.2   EVALUATION METRICS

**Automatic Evaluation.**   Here we briefly introduce the formulation of Bleu-2 and Rouge-L.

*Bleu-2* (B-2) (Papineni et al., 2002) first computes the geometric average of the modified $n$-gram precisions, $p_n$, using $n$-grams up to length $N$ and positive weights $w_n$ summing to one. Next, let $c$ be the length of the prediction and $r$ be the reference length. The BP and Bleu-2 are computed as follows:

$$\text{BP} = \begin{cases} 1 & \text{if } c > r \\ e^{(1-r/c)} & \text{if } c \leq r \end{cases}. \tag{22}$$

$$\text{Bleu} = \text{BP} \cdot \exp\left(\sum_{n=1}^{N} w_n \log p_n\right). \tag{23}$$

*Rouge-L (R-L)* (Lin, 2004) proposes LCS-based F-measure to estimate the similarity between two summaries $X$ of length $m$ and $Y$ of length $n$, assuming $X$ is a reference summary sentence and $Y$ is a candidate summary sentence, as follows:

$$R_{lcs} = \frac{LCS(X,Y)}{m}$$
$$P_{lcs} = \frac{LCS(X,Y)}{n} \tag{24}$$
$$F_{lcs} = \frac{(1 + \beta^2) R_{lcs} P_{lcs}}{R_{lcs} + \beta^2 P_{lcs}}$$

Where $LCS(X, Y)$ is the length of a longest common subsequence of $X$ and $Y$, and $\beta = P_{lcs}/R_{\text{lcs}}$ when $\partial F_{lcs}/\partial R_{lcs} = \partial F_{lcs}/\partial P_{lcs}$. In DUC, $\beta$ is set to a very big number ($\rightarrow \infty$). Therefore, the LCS-based F-measure, i.e. Equation 24, is Rouge-L.

Table 6: Critical training configurations of HdLM on different tasks and datasets. HTC denotes hierarchical text classification; CgG denotes classification-guided generation; HTG denotes hierarchical text generation.

| Method | HTC | | CgG | HTG | |
|--------|-----|-----|-----|-----|-----|
| | WoS | DBPedia | ESConv | ToMI | BigToM |
| k | [25] | [20, 30] | 28 | 24 | 24 |
| lr | 5.0e-6 | 5.0e-7 | 1.0e-6 | 1.0e-6 | 1.0e-6 |
| loss weights | [2,1] | [3,2,1] | [1,3] | [4,1] | [4,1] |
| epoch | 10 | 10 | 4 | 2 | 2 |

*CIDEr.* The $CIDEr_n$ Vedantam et al. (2015) score for $n$-grams of length $n$ is computed using the average cosine similarity between the candidate sentence and the reference sentences, which accounts for both precision and recall:

$$CIDEr_n(c_i, S_i) = \frac{1}{m} \sum_j \frac{\mathbf{g^n}(c_i) \cdot \mathbf{g^n}(s_{ij})}{\|\mathbf{g^n}(c_i)\| \|\mathbf{g^n}(s_{ij})\|}, \tag{25}$$

where $\mathbf{g^n}(c_i)$ is a vector formed by $g_k(c_i)$ corresponding to all $n$-grams of length $n$ and $\|\mathbf{g^n}(c_i)\|$ is the magnitude of the vector $\mathbf{g^n}(c_i)$. Similarly for $\mathbf{g^n}(s_{ij})$.

Higher order (longer) $n$-grams are used to capture grammatical properties as well as richer semantics. (Vedantam et al., 2015) combine the scores from $n$-grams of varying lengths as follows:

$$CIDEr(c_i, S_i) = \sum_{n=1}^{N} w_n CIDEr_n(c_i, S_i), \tag{26}$$

Empirically, Vedantam et al. (2015) found that uniform weights $w_n = 1/N$ work the best. So, We also use $N = 4$.

## C.3 KEY PARAMETERS

Table 6 compares the dataset-wise configurations, including the learning rate, decoding layer indexes and their loss weights.

## C.4 BASELINES

Further details of the baselines are provided:

(1) Direct: directly infer the LLM, with the same context.
(2) Direct-Refine: a straightforward refinement method in which the model revises its initial response to incorporate emotional support considerations.
(3) Self-Refine: a method (Madaan et al., 2023) initiates by generating feedback emphasizing emotional support from the initial response, then refining the response based on this feedback.
(4) CoT: uses the Chain-To-Thought prompt (Wei et al., 2022), which first generate the seeker's *emotion*, which then guides the generation of strategy and response.
(5) FSM: the finite state machine (Wang et al., 2024) with finite sets of states and state-transitions triggered by inputs, and associated discrete actions.

## D MORE RESULTS

### D.1 DETAILED PROOF

Here we provide the detailed proof that $\mathcal{F}_{\text{infer}}^{\text{baseline}} > \mathcal{F}_{\text{infer}}^{\text{HdLM}}$:

$$\mathcal{F}_{\text{infer}}^{\text{baseline}} = fK \sum_{j=1}^{L_1+L_2} (L + j - 1) \tag{27}$$

$$\mathcal{F}_{\text{infer}}^{\text{HdLM}} = fk_1 \sum_{j=1}^{L_1} (L + j - 1) + f(K - k_1) \sum_{j=1}^{L_2} (L + L_1 + j - 1) \tag{28}$$

$$\sum_{j=1}^{L_1+L_2} (L + j - 1) = \sum_{j=1}^{L_1} (L + j - 1) + \sum_{j=1}^{L_2} (L + L_1 + j - 1) \tag{29}$$

Rewriting the Baseline Formula

$$\mathcal{F}_{\text{infer}}^{\text{baseline}} = fK \left( \sum_{j=1}^{L_1} (L + j - 1) + \sum_{j=1}^{L_2} (L + L_1 + j - 1) \right) \tag{30}$$

$$\mathcal{F}_{\text{infer}}^{\text{baseline}} - \mathcal{F}_{\text{infer}}^{\text{HdLM}}$$

$$= fK \left( \sum_{j=1}^{L_1} (L + j - 1) + \sum_{j=1}^{L_2} (L + L_1 + j - 1) \right)$$

$$- \left[ fk_1 \sum_{j=1}^{L_1} (L + j - 1) + f(K - k_1) \sum_{j=1}^{L_2} (L + L_1 + j - 1) \right]$$

$$= f(K - k_1) \sum_{j=1}^{L_1} (L + j - 1) + fk_1 \sum_{j=1}^{L_2} (L + L_1 + j - 1)$$

$$> 0$$

### D.2 LOSS ANALYSIS

Figure 5 (left) shows the loss curves of WoS. Because the $k_1$-th layer is not originally designed to generate text, its $\mathcal{L}_1$ is large at the beginning of training. Nevertheless, it converges to a low value, indicating the $k_1$-th layer is successfully learned to generate the thought. Furthermore, $\mathcal{L}_2$ also decays to a lower value, since the final layer adapts to decode grounded by both the query and the output of $k_1$-th layer.

Figure 5 (right) shows the averaged stable $\mathcal{L}_1$ on different $k$. As $k$ becomes larger, the representing capability of $\mathcal{M}_{0:k_1}$ increases such that higher proficiency can be reached.

Figure 6 further exhibits different loss curves with respect to different $k_1$ (left), and the enlarged part of the starting 50 steps (right). Besides the stable values, the transient behavior of $\mathcal{L}_1$ also differs from $k_1$, and a larger $k_1$ means faster convergence. Nevertheless, one also needs to notice that although a larger $k_1$ can help the learning of the $k_1$-th layer, the net depth between the $k_1$-th layer and the final layer becomes smaller, therefore might hurt the final response quality.

### D.3 MORE RESULTS OF CGG

**Human Evaluation.** Table 7 presents human evaluation results. HdLM achieves the highest scores in almost all aspects, which verifies the automatic evaluation results.

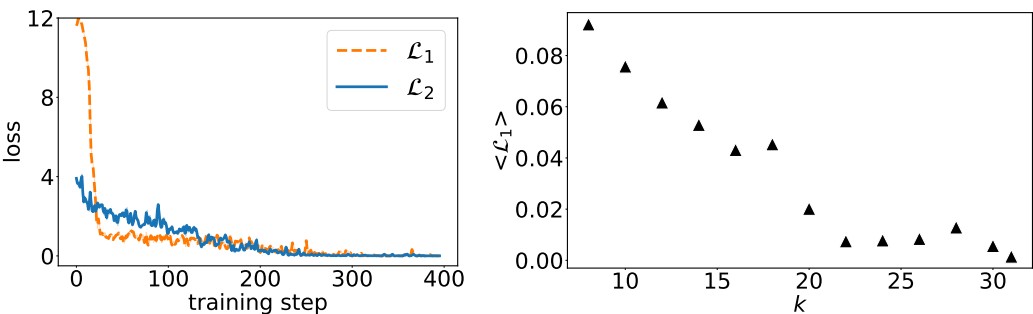

Figure 5: Loss analysis of HdLM on ESConv with depth $D = 2$. Left: loss curves of the first level $\mathcal{L}_1$ and the final level $\mathcal{L}_2$. Right: the final averaged $\mathcal{L}_1$ with respect to different choices of $k_1$.

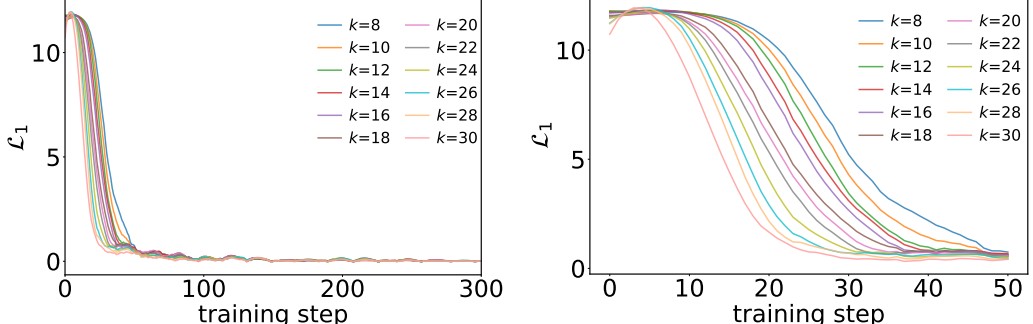

Figure 6: Loss curves on ESConv with different choices of $k_1$ (Left) and the enlarged exhibition of the beginning 50 steps (Right).

| Method | Fluency | Emotion | Acceptance | Effectiveness | Sensitivity | Alignment | Satisfaction |
|---|---|---|---|---|---|---|---|
| Direct | 2.95±1.41 | 3.00±1.34 | 2.60±1.15 | 2.40±0.92 | 2.70±1.08 | 2.70±1.08 | 2.60±1.41 |
| + Refine | 3.09±1.25 | 3.09±1.16 | 2.73±1.22 | 2.91±1.41 | 2.91±1.23 | 2.82±1.25 | 2.84±1.40 |
| + Self-Refine | 3.10±1.29 | 3.15±1.38 | 2.80±1.19 | 2.70±1.14 | 2.90±1.03 | 2.80±1.16 | 2.80±1.20 |
| + CoT | 3.08±1.02 | 3.08±1.29 | 2.83±1.27 | 2.67±1.06 | 3.00±1.27 | 2.83±1.13 | 2.83±1.10 |
| + FSM | 3.30±1.32 | 3.35±1.38 | 2.90±1.17 | 2.90±1.03 | 3.00±1.46 | 2.90±1.15 | 2.93±1.19 |
| + SFT | 3.15±1.44 | 3.40±1.30 | 2.70±1.19 | 2.70±1.20 | 2.90±1.24 | 3.30±1.32 | 2.90±1.32 |
| + CoT + SFT | 3.67±1.21 | 3.61±1.17 | 3.22±1.25 | 3.67±1.26 | 3.56±1.13 | 3.35±1.39 | 3.45±1.31 |
| + FSM + SFT | 3.80±1.26 | 3.55±1.16 | 3.40±1.21 | 3.70±1.14 | 3.80±1.06 | 3.70±1.04 | 3.65±1.19 |
| **+ HdLM** | **3.84**±1.10 | **3.67**±1.47 | **4.07**±0.87 | **3.96**±0.99 | **3.83**±1.14 | **3.85**±1.09 | **3.86**±1.12 |

Table 7: Average human scores (with standard deviations) of response quality on ESconv.

Table 8: Win-tie-lose rates (%) of GPT-4o evaluation of HdLM and FSM compared to SFT, all of which are fine-tuned on LLama3-8B-Instruct.

| vs SFT | win ↑ | tie | lose ↓ |
|--------|-------|------|--------|
| FSM | 59.5 | 11.1 | 29.2 |
| **HdLM** | **61.5** | 11.7 | **26.8** |

**Results of LLM-as-a-Judge.**   Table 8 also provides GPT-4o evaluation results of HdLM and FSM, with the pairwise comparison to SFT. HdLM achieves higher win rate and lower lose rate.

### D.4   ToM Tests

We then conduct another series of theory-of-mind (ToM) tests, with the Sally-Anne false belief tests (ToMI (Le et al., 2019) and BigToM (Gandhi et al., 2024)) as ID tests, and the Vignette-based test (Binz & Schulz, 2023) as the ODD test.

For ToM tests, we further introduce more domain-specific approaches, including SimToM (Alex Wilf, 2022) and Quiet-STaR (Eric Zelikman, 2022b).

Table 9: Scores (in percentage) of ToMI, BigToM (ID) and vignetted-based tests (OOD).

| Base | Method | HTG ($\mathcal{G}_{rationale} \to \mathcal{G}_{answer}$) | | |
|------|--------|-----------|--------------|----------------|
| | | ToMI (ID) | BigToM (ID) | Vignette (OOD) |
| *models with much larger size:* | | | | |
| GPT-3 | Direct | - | - | 37.5[⋆] |
| GPT-4 | Direct | 92.5[▲] | 66.5[▲] | 46.9 |
| GPT-4 | CoT | 95.5[▲] | 74.4[▲] | - |
| GPT-4 | SimTom (Alex Wilf, 2022) | 95.0[▲] | 87.8[▲] | - |
| *models with similar size:* | | | | |
| Mistral-7B | Direct | - | - | 40.2 |
| Mistral-7B | Quiet-STaR (Eric Zelikman, 2022b) | - | - | 11.1 |
| Llama3-8B | Direct | 22.2 | 71.3 | 23.8 |
| Llama3-8B | SFT | 43.2 | 77.7 | - |
| Llama3-8B | **HdLM (ours)** | **98.2** | **99.4** | **48.3** |

▲: results from Alex Wilf (2022); ⋆: result from Binz & Schulz (2023).

HdLM even outperforms GPT-3 and GPT-4 (Team, 2024) that have a much larger size. Also, Quiet-STaR (based Mistral-7B-instruct), although it also has an internal thinking mechanism, fails to capture the Vignette-based scenario, since it is more focused on math reasoning.

### D.5   Good Cases

**Case of HTC.**   Detailed cases of WoS and DBpedia are in Table 10.

**Case of CgG.**   Detailed cases of ESConv and EmpatheticDialogues are in Table 11.

**Case of HTG.**   Typical cases of AQuA and CommonSenseQA are in Table 12; cases of ToMI and BigToM are in Table 13; Table 14 provides a Vignette-based example.

**Open-domain Case.**   Table 15 provides an open-domain planning case, in which HdLM automatically use the $k_1$ layer to make coarse-grain plans, plan in an abstraction level, helping organize the final response.

Table 10: Typical cases of HTC.

| | Case of WoS |
|---|---|
| $q$ | Abstract: The ability to engineer cells to express a protein of interest in an inducible manner and stably for a long period is a valuable tool in molecular biology and also one that holds promise for regenerative medicine in the future. CCN proteins have been suggested to be involved in tissue regeneration. In this chapter, we describe an in vitro method for stable and inducible expression of CCN protein in a chondroprogenitor cell line and in chondrocytes in primary culture that does not involve the use of any viral vector. Keywords:PiggyBac; Transposon; Stable; Expression; CCN. Domain List: 0: 'CS', 1: 'ECE', 2: 'Psychology', 3: 'MAE', 4: 'Civil', 5: 'Medical', 6: 'Biochemistry' |
| $r_1$ | Domain ID: 6

Area List: 0: 'Molecular biology', 1: 'Cell biology', 2: 'Human Metabolism', 3: 'Immunology', 4: 'Genetics', 5: 'Enzymology', 6: 'Polymerase chain reaction', 7: 'Northern blotting', 8: 'Southern blotting' |
| $r_2$ | Area ID: 0 |
| | Case of DBpedia |
| $q$ | WIKI NAME: Dusky lark
WIKI CONTENT: The dusky lark (Pinarocorys nigricans) is a species of lark in the Alaudidae family. It is found in Angola, Botswana, Democratic Republic of the Congo, Malawi, Mozambique, Namibia, South Africa, Swaziland, Tanzania, Zambia, and Zimbabwe. Its natural habitats are dry savannah and subtropical or tropical dry lowland grassland.

Top-Level Labels: 0: 'Agent', 1: 'Device', 2: 'Event', 3: 'Place', 4: 'Species', 5: 'SportsSeason', 6: 'TopicalConcept', 7: 'UnitOfWork', 8: 'Work' |
| $r_1$ | Top-Level ID: 4

Mid-Level Labels: 0: 'Animal', 1: 'Eukaryote', 2: 'FloweringPlant', 3: 'Horse', 4: 'Plant' |
| $r_2$ | Mid-Level ID: 0

Bottom-Level Labels: 0: 'Amphibian', 1: 'Arachnid', 2: 'Bird', 3: 'Crustacean', 4: 'Fish', 5: 'Insect', 6: 'Mollusca', 7: 'Reptile' |
| $r_3$ | Bottom-Level: 2 |

### D.6 COMPUTATION RESULTS

Figure 7 shows the training FLOPs recorded during our experiments. The input sequence length is fixed to 256 while $L_1$ is always kept the same with $L_2$. The linear relationship of FLOPs with both $k$ and $L_2$ are consistent with our theoretical derivations.

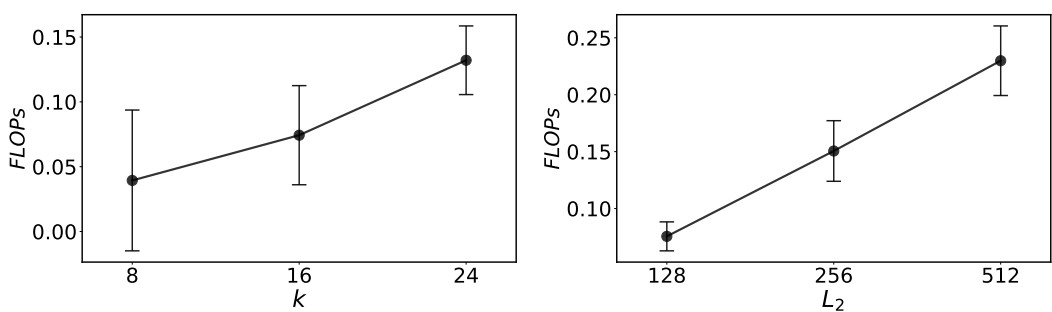

Figure 7: Computation saving of trainings as functions of $k$ (left) and $L_2$ (right).

Table 11: Typical cases of CgG.

| | Case of EmpatheticDialogues |
|---|---|
| $q$ | History:
user: I feel so lonely because I have no friends.
assistant: I'm sorry to hear that. Have you tried joining clubs or hanging out with people from work?
user: I haven't. I'm too shy and I am scared of what people will say.

Emotion candidates:
0: 'anger', 1: 'anxiety', 2: 'depression', 3: 'disgust', 4: 'fear', 5: 'guilt', 6: 'jealousy', 7: 'nervousness', 8: 'pain', 9: 'sadness', 10: 'shame'. |
| $r_1$ | Emotion ID: 9

Strategy candidates: 0: 'Question', 1: 'Others', 2: 'Providing Suggestions', 3: 'Affirmation and Reassurance', 4: 'Self-disclosure', 5: 'Reflection of feelings', 6: 'Information', 7: 'Restatement or Paraphrasing' |
| $r_2$ | Strategy ID: 0 |
| $r_3$ | I understand that it can be scary to put yourself out there, but I think you'll find that most people are friendly and willing to help. |
| | Case of ESConv |
| $q$ | History:
supporter: Hi, how are you feeling tonight? Is there anything I can help you with?
seeker: Hi.
seeker: Feeling upset.
supporter: I'm sorry to hear that. Can I talk you through what you're experiencing?
seeker: I am sad because of my love.
supporter: That's frustrating to hear. What is your love?
seeker: It's gone, she left me.

Emotion candidates:
0: 'anger', 1: 'anxiety', 2: 'depression', 3: 'disgust', 4: 'fear', 5: 'guilt', 6: 'jealousy', 7: 'nervousness', 8: 'pain', 9: 'sadness', 10: 'shame'. |
| $r_1$ | Emotion ID: 9

Strategy candidates: 0: 'Question', 1: 'Others', 2: 'Providing Suggestions', 3: 'Affirmation and Reassurance', 4: 'Self-disclosure', 5: 'Reflection of feelings', 6: 'Information', 7: 'Restatement or Paraphrasing' |
| $r_2$ | Strategy ID: 3 |
| $r_3$ | I would suggest trying to get a tutor, or even a study group. I personally found that it helped me a lot. I also tried to make a schedule for myself, so I could keep track of my assignments and due dates. |

## D.7 FURTHER SENSITIVITY STUDIES

**Analysis on loss weights.** Figure 8 visualizes different performances of WoS and DBpedia with different loss weights. From the optimal choices, we suppose the higher levels of responses need a higher weight since their layer depths are smaller than lower levels.

## D.8 MODEL PARAMETER ANALYSIS

Figure 9 shows the model parameter differences between SFT and the original base model, while Figure 10 shows the model parameter differences between HdLM and the base model.

It is evident that after finetuning, the base model and SFT exhibit differences in parameters across all layers, although the magnitude of these differences is not uniformly distributed across layers. In contrast, parameter differences between HdLM and the base model are primarily concentrated within the first 24 layers (here we select $k = 24$). This indicates that our finetuning of the intermediate layers has been effective, successfully altering the output logic of the intermediate layers.

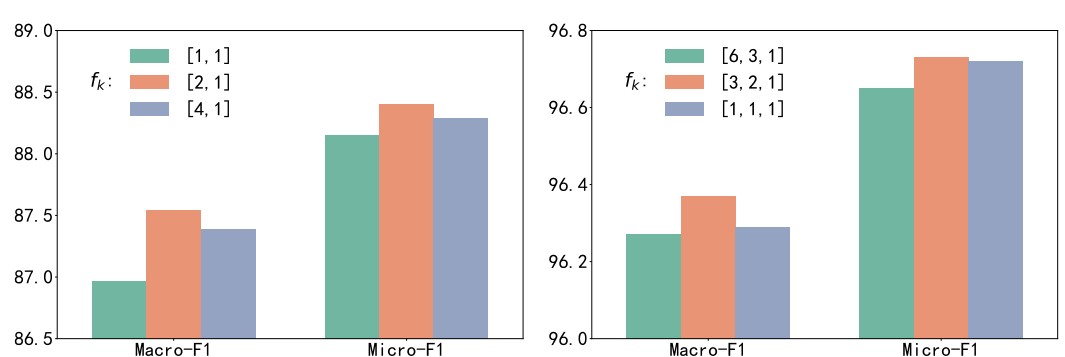

Figure 8: Sensitivity studies of loss weights ($f_k$). Left: WoS; Right: DBpedia.

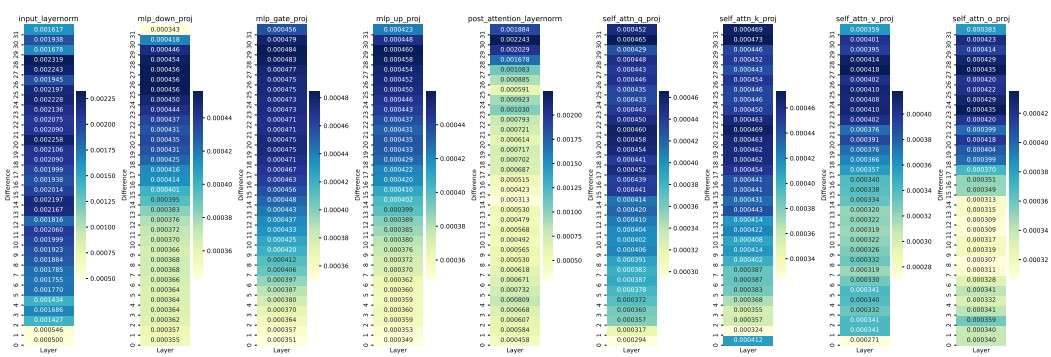

Figure 9: Visualization of model parameter differentiation, between Llama3-8B-Instruc and SFT, with respect to all layers.

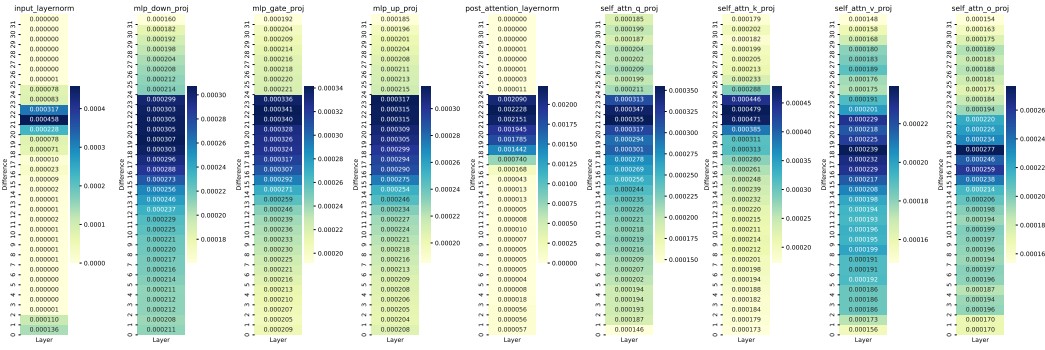

Figure 10: Visualization of model parameter differentiation, between Llama3-8B-Instruct and HdLM, with respect to all layers.

Table 12: Typical Cases of Reasoning Benchmarks.

| | Case of AQuA |
|---|---|
| $q$ | A sales person gets a 10% commission on each sale he makes. How many sales of $250 each must he make in order to reach a salary of at least $1000? |
| $r_1$ | As original question is 10% of each sale = $250*10/100 = $25
for 1000 $25*40 =1000
ans is 40 |
| $r_2$ | D |
| | Case of CommonSenseQA |
| $q$ | Question: Sammy wanted to go to where the people were. Where might he go?
Identify the main subject in the question and recall relevant commonsense knowledge about it. |
| $r_1$ | The main subject of the question is "Sammy wanted to go to where the people were." From this, I can infer that Sammy wants to go to a place with a large number of people. This is a common sense assumption, as people tend to congregate in areas with a high population density.
With this in mind, I think Sammy might go to:
# Answer: B: populated areas |
| $r_2$ | B |

Table 13: Typical cases of Sally-Anne false belief experiments. Text in yellow is the role assigned to the LLM. Text in blue indicates HdLM's consideration. Text in green is the final result obtained HdLM.

| | | |
|---|---|---|
| ToMI | $q$ | Story:
**1** Amelia entered the office. **2** Sophia entered the office.
**3** The sweatshirt is in the container.**4** The container is in the office.
**5** Amelia exited the office. **6** Logan entered the office.
**7** Sophia moved the sweatshirt to the bottle. **8** The bottle is in the office.
You are Amelia . Based on the above information, answer the following question:
Where will Amelia look for the sweatshirt? |
| | $r_1$ | I am Amelia .
Based on the above information, from my point of view, what I know is:
**1** Amelia entered the office. **2** Sophia entered the office.
**3** The sweatshirt is in the container. **4** The container is in the office.
**5** Amelia exited the office. |
| | $r_2$ | container |
| BigToM | $q$ | Story:
Fatima is an astronaut aboard a space station, conducting scientific experiments. She needs to retrieve a sample from the storage area to conduct a crucial test. Fatima sees that the sample is stored in a sealed container. A small meteoroid hits the space station, puncturing the container and exposing the sample to the vacuum of space. Fatima does not notice the damage caused by the meteoroid.
You are Fatima . Based on the above information, answer the following question:
What will Fatima do? Choose one of the following:
a) Fatima will retrieve the sealed container and proceed with the experiment.
b) Fatima will report the damaged sample to mission control and ask for advice.
Answer the questions based on the context. Keep your answer concise, a few words are enough, maximum of one
sentence. Answer as 'Answer:¡option¿)¡answer¿' |
| | $r_1$ | I am Fatima . Based on the above information, from my point of view, what I know is:
Fatima is an astronaut aboard a space station, conducting scientific experiments. She needs to retrieve a sample from the storage area to conduct a crucial test. Fatima sees that the sample is stored in a sealed container. |
| | $r_2$ | Answer: a) Fatima will retrieve the sealed container and proceed with the experiment. |

Table 14: Typical cases of Vignette-based experiments.

| | Case of Vignette |
|---|---|
| $q$ | Imagine a box that makes a sound if certain objects are placed on it. When I put a red object on it, the box made a sound. When I put a green object on it, the box made a sound. When I put a blue object on it, the box did not make a sound. When I put a yellow object on it, the box did not make a sound.

Q: I put a green and a yellow object on the box and it made a sound. Would it still have made a sound if I had not put the yellow object on it? |
| $r_1$ | Alright, I need to figure out whether the box made a sound when I put both green and yellow objects onto it. Let's break this down step by step.

First, let's understand the given information:
- The box makes a sound if certain object is placed on it.
- Every time I put a red object on the box, it made a sound.
- On the other hand, when I put a green object on the box, it made a sound.
- In addition, when I put a blue object on the box, it did not make a sound.
- On the other hand, when I put a yellow object on the box, it did not make a sound.

Given that: - When I put a green object on the box, it made a sound.
- When I put a yellow object on the box, it did not make a sound.

Now, I'm going to place two different types of objects into the box:
1. **Green object**: Can make a sound because it's a green color.
2. **Yellow object**: Can't make a sound because it's yellow color.

If I have only one type of object, say red, the problem is straightforward:
- If I put a green object on the box, it made a sound.
- If I put a yellow object on the box, it did not make a sound.

But here, we have two types of objects:
- Green and yellow objects.

So, to sum up, the box did make a sound when I put both green and yellow objects into it. If I had left the yellow object behind, the box would have made a sound, but in this case, the color of the yellow object is not relevant to the statement about the box making a sound when the green object was placed there. |
| $r_2$ | Yes, if you hadn't placed the yellow object on the box, the box would still have made a sound when the green object was placed on it. |

## D.9 ABLATION ON METHODOLOGY COMPONENTS

There are also other choices of hierarchical paradigms. To consolidate our conclusion, we also explore these paradigms on HTC tasks:

(a) *w/ head*: implement the classification head on the intermediate layers and the final layer, and use them to study the classification tasks.

(b) *cascade*: use two separate LLMs, finetuned and inferenced by $r_1$ and $r_2$, respectively.

(c) *multi-hop*: inference a mix-finetuned LLM twice, with $r_1$ and $r_2$ inferenced sequentially.

(d) *sft@K*: use the final layer to decode both $r_1$ and $r_2$.

Table 16 shows the ablation result on WoS. HdLM still holds the best results which indicates our framework is rigorous. Furthermore, we also compare HdLM to two imaginary solutions

(e) *true-1*: use the ground truth $r_1$ and only conduct the $r_2$ classification (from the full $r_2$ list).

(f) *true-1 & true-2 candidates*: use the ground truth $r_1$ as well as ground truth $r_2$ candidates.

Both of which are apparently not applicable in practice. Even so, HdLM's performance is still close to theirs, indicating a high-level comprehension of label hierarchy.

Table 15: An open-domain planning case of HdLM. Text in blue indicates HdLM's consideration.

| Query |
| --- |
| Write a C program that prints "Hello, World!" to the console. Explain the code structure and key components. |

**Thought**

To tackle this task, I need to:
1. Recall the basic structure of a C program
2. Explain each component of the code
3. Provide a clear explanation of the code
I'll start by outlining the basic structure, then break down each component, and finally explain how they work together.
1. Brief introduction to reinforcement learning
2. Key principles
3. Comparison with other machine learning models
4. Applications and advantages
5. Step-by-step process for implementation
Upon reflection, I realize I should also mention
The importance of the function (e.g., "The algorithm is important") in the explanation. Additionally, I should clarify that the basic structure is not just a basic class or Kecid's algorithm, but a simple C program for finding the number of food classes to check for the final goal.

**Answer**

Here's a C program that prints "Hello, World!" to the console:

```c
#include <stdio.h>

int main() {
    printf("Hello, World!\n");
    return 0;
}
```

Let's break down the code structure and key components:

1. '#include <stdio.h>': This line includes the standard input/output header file, which provides functions like 'printf()'.

2. 'int main()': This declares the main function, which is the entry point of the program.

3. ' ... ': These braces enclose the body of the main function.

4. 'printf("Hello, World!\n");': This line uses the 'printf()' function to print the string "Hello, World!" to the console. The '\n' at the end adds a newline character for proper formatting.

5. 'return 0;': This line returns 0 from the main function, indicating successful execution of the program.

This program demonstrates basic C syntax and usage of the 'printf()' function. It's a fundamental example often used to introduce new programmers to the language.

Some related concepts you might find interesting are:
1. Command-line arguments in C programs
2. Error handling in C
3. Using functions for modular programming

Would you like me to explain any part of the code in more detail?

Table 16: Ablation on WoS (HTC task).

| Metrics | Ablation | | | | | Not applicable practically | |
|---------|---------|---------|-----------|--------|-----------|--------|----------------------------|
| | w/ head | cascade | multi-hop | sft@$K$ | **HdLM** | true-1 | true-1 & true-2 candidates |
| MiF1 | 82.38 | 87.92 | 64.51 | 86.64 | **88.40** | 89.44 | 92.78 |
| MaF1 | 37.24 | 86.84 | 68.62 | 85.86 | **88.54** | 88.61 | 89.21 |

## E  LIMITATION AND FUTURE DIRECTIONS

**Possibility of Training from Scratch.**   Due to resource and limit, we choose to finetune from a pretrained LLM. However, we speculate that the HdLM approach can also work in an end-to-end training paradigm, and one may expect more capabilities to emerge during this progress.

**Automatic selection of $k$.**   In this work, we show both theoretical guidance and empirical conclusions of $k$ selection. This information may need task-specific training. A self-adaptive paradigm with automatic $k$ determination would be highly attractive.

**Other hierarchical tasks.**   There might be even more complicated hierarchical tasks. For example, the generative classification ($\mathcal{G} \rightarrow \mathcal{C}$) and 'think, classify and act' ($\mathcal{G} \rightarrow \mathcal{C} \rightarrow \mathcal{G}$). We will explore these possibilities in the future.

