# OpenReview forum: "Decoding Layer by Layer: Uncovering Hierarchical Reasoning in Language Models"
_ICLR.cc/2026/Conference — Submitted to ICLR 2026_

### Official Review · Reviewer_7VJP · 2025-10-20

**Soundness:** 3
**Presentation:** 2
**Contribution:** 3
**Rating:** 4
**Confidence:** 4

**Summary:**

The paper discusses work on optimizing LLMs to exploit hierarchical reasoning through layer-specific probing. The call this approach HdLM. The authors claim that hierarchical reasoning is grounded from how humans reason sequentially through conceptual abstractions and sequential determination. The authors explore this setup across different tasks including hierarchical text classification (HTC), classification-guided generation (CG), and hierarchical text generation (HTC). Results show that HdLM obtains better performance across BERT-based and retrieval-based methods for HTC from WoS and DBPedia benchmarks, better performance than prompting-based optimizations like vanilla CoT, SelfRefine, direct, and FSM for CgC from ESConv and EmpatheticDialogues, and better performance against commercial models for HTG. The authors also provide ablation experiments on effects with model scale and layer selection which also supports the effectivity of the approach.

**Strengths:**

The paper is readable and tackles and interesting and timely work on probing the hierarchical reasoning for LLMs and its effectiveness across reasoning-based tasks. This is an important research topic where this work can shed light on specific knowledge encoded in an LLM’s layers and how this can be exploited for better reasoning capabilities. Technicality-wise, I appreciate the thorough ablations such as the layer-wise effects, model scale performed by the authors to support effectiveness of their approach. Showing that the method works across these variables is an advantage.

**Weaknesses:**

The paper can benefit from a better framing and motivation. The authors mention drawing motivation from how humans use hierarchical reasoning but the supporting information provided are limited (lines 42-48). What specific tasks or examples can you provide that draws from both the “multi-timescale abstraction” and “sequential determination”? The figure 1 should be anchored on these two aspects. Likewise, the authors should reference interdisciplinary literature from cognition and psychology (which I believe is lacking from the current paper) since the work claims to be inspired by human paradigm of sequential reasoning and abstraction.

The related work at the very end is weak and uninsightful despite the rich literature in hierarchical LLMs. Please improve the depth of the search and connect literature on effectivity of probing LLM layers across tasks, efficiency of hierarchical transformer models, and applications of hierarchical encoder/decoder layers. As mentioned above, the paper seems to be missing a crucial connection with cognitive science literature if the authors are intent on grounding the motivation to the way humans reason hierarchically.

One main issue is that the discussion and results seem very surface-level and limited in insights. For example, one of the most important part is analyze which layer works with respect to task, however, the paper does not convincingly discusses **why** the these specific layers work and whether these changes for specific tasks, optimization setup, model architecture or model scale. What’s with earlier layers that seem to be very poor across both tasks? Does this mean earlier layers do not encode information? How about mid to later layers? Does converging performance mean they encode semantic information better than earlier or later layers? How do these tie to the type of knowledge required by tasks  WoS and ESConv? All these should be discussed succinctly in the paper but are currently missing.

One novel improvement that can potentially strengthen the paper technically is an automatic search-based algorithm for selecting the minimum number of layers that can still reach the highest performance. In 4.4, the authors mention selecting layers as much as 25 and 28 for each task but this seems to be excessive and unintuitive. This way, you do not have to go through all layers and maybe start intuitive such as with middle layers based from ablation experiments.

Minor: “Motivated by both mechanisms, models with hierarchical reasoning capabilities have recently been explored, either implementing on (i) (Barrault et al., 2024; Wang et al., 2025) or (ii) (Yang et al., 2024; Cai et al., 2024).” - On what? This is a hanging sentence and leaves readers confused. Please provide succinct supporting information from the cited references.

Minor: Please fix citations: “(Jason Wei, 2022)”, “Llama (AI@Meta, 2024)”, “ (Eric Zelikman, 2022b)”

**Questions:**

In Figure 1, the figures show exact same outputs between conventional LM vs. HdLM. While I do get the idea that the authors are relaying, I was confused at first why they are the same and not slightly different since you are getting outputs from various layers for HdLM. The authors should revise this figure just to ensure that readers will not assume that the output from a vanilla LM is exactly the same as HdLM.

The layer-wise analysis shows almost no difference in performance for both tasks across mid to later layers for WoS. Do you think the difference by selection layer 25 as the best one compared to layers 10 to 31 statistically significant? How about for the layers for ESConv task?

---

> ### Author Response · Authors · 2025-11-25
>
> > In Figure 1, the figures show exact same outputs between conventional LM vs. HdLM
>
> We sincerely appreciate your insightful feedback. We have revised the examples in Figure 1 to more clearly illustrate our core methodology.
>
> > The layer-wise analysis shows almost no difference in performance for both tasks across mid to later layers for WoS. Do you think the difference by selection layer 25 as the best one compared to layers 10 to 31 statistically significant? How about for the layers for ESConv task?
>
> We are sorry that the selection principle for layer decomposition was not sufficiently clarified. As studied by [1], middle layers encode valuable  representations for downstream tasks, and the 32-layer LLaMA-8B model can be partitioned into three segment:
> - region of **shared layers (1-9)**
> - region of **transition layers (10-15)**
> - region of **refinement layers (16-32)**
> therefore, for high-level decoding purpose, the optimal layer selection should fall within the refinement region (16 ≤ k ≤ 32).
>
> Futhermore, [2] observes that LLM first undergoes a *`compression valley`* where entropy decreases until k = 25, followed by entropy increase; [3] identifies k = 25 as the intersection point between the *concept generation* and *token generation* phases. These findings collectively justify narrowing the optimal k range to around **25 ≤ k ≤ 30**.
>
> The above narrowed region allows us to conduct empirical analysis to finally determine the optimal k with much less costs. Our empirical findings in Figure 4 (Section 4.4) reveals that: HTC (WoS) accuracy remains robust across k within this region; while CgG exhibits a clear trade-off requiring more layers for generation (e.g., k = 28). These observations lead us to conclude that **classification tasks require fewer layers than generation tasks** (mentioned in Section 4.4). As a result, **k of CgG and HTG should be sligntly larger than k of HTC**, which finally implies the formal settings: **k=25 for WoS and k=28 for ESConv**, which align precisely with the theoretical guidance from [1]–[3]. We will emphasize this alignment in the revised manuscript.
>
> > What specific tasks or examples can you provide that draws from both the “multi-timescale abstraction” and “sequential determination”? The figure 1 should be anchored on these two aspects.
>
> We have revised Figure 1 to explicitly illustrate two mechanisms. Updated examples are as follows:
>
> 1. *“multi-timescale abstraction”*: In emotional support dialogues, the model first determines a high-level strategy (e.g., asking, concluding, or introducing a new topic), then generates granular responses aligned with that strategy (e.g., "How about your feelings?" following the asking strategy).
>
> 2. *“sequential determination”*: In chain-of-thought reasoning, the model processes intermediate steps sequentially before arriving at the final answer.
>
> > the authors should reference interdisciplinary literature from cognition and psychology (which I believe is lacking from the current paper) since the work claims to be inspired by human paradigm of sequential reasoning and abstraction.
>
> We have incorporated the relevant references into Section 1 with appropriate contextual discussions:
>
> - A hierarchy of intrinsic timescales across primate cortex. Nature neuroscience 2014.
> - Large-scale gradients in human cortical organization. Trends in cognitive sciences 2018
> - Intrinsic timescales in the visual cortex change with selective attention and reflect spatial connectivity. Nature communications 2023
>
> > The related work at the very end is weak and uninsightful despite the rich literature in hierarchical LLMs.
>
> We have also expand our related work to enhance our motivations.
>
> > One novel improvement that can potentially strengthen the paper technically is an automatic search-based algorithm for selecting the minimum number of layers that can still reach the highest performance.
>
> We sincerely appreciate the reviewer's insightful suggestion and agree that this represents a promising direction for future work. Specifically, we will implement a layer-wise switcher leveraging speculative decoding at each layer (integrated with LayerSkip [4]): when the current layer's decoding is correct, we retain the intermediate solution (as in our current framework); otherwise, we skip intermediate decoding and use the intermediate latent solely to guide subsequent layer generation, transitioning smoothly to standard LLM operation. These enhancements will be explicitly highlighted in the revised manuscript. We deeply value the reviewer's constructive feedback.
>
> ### References:
>
> [1] Layer by Layer: Uncovering Where Multi-Task Learning Happens in Instruction-Tuned Large Language Models, *EMNLP 2024*
>
> [2] Layer by Layer: Uncovering Hidden Representations in Language Models, *ICML 2025*
>
> [3] Multilingual Contrastive Decoding via Language-Agnostic Layers Skipping, *EMNLP 2024*
>
> [4] LayerSkip: Enabling Early Exit Inference and Self-Speculative Decoding, *ACL 2024*

---

> > ### Comment · Reviewer_7VJP · 2025-11-27
> >
> > This is to confirm that I have read the authors' response to my review as well as my co-reviewers' feedback and corresponding responses by the authors.
> >
> > > Futhermore, [2] observes that LLM first undergoes a compression valley where entropy decreases until D = 25, followed by entropy increase; [3] identifies D = 25 as the intersection point between the concept generation and token generation phases. These findings collectively justify narrowing the optimal D range to around 25 ≤ D ≤ 30.
> >
> > Does this mean it's not plausible or practical for HdLM or any technique at that to use layers outside of 25 ≤ D ≤ 30? If so, this would make using HdLM quite limiting since there's already a predefined subset of layers that work from previous literature (Skean et al, ICML 2025) that are effective for layer-wise experiments. Kindly clarify this please.

---

> > > ### Author Response · Authors · 2025-12-02
> > >
> > > > Does this mean it's not plausible or practical for HdLM or any technique at that to use layers outside of 25 ≤ D ≤ 30? If so, this would make using HdLM quite limiting since there's already a predefined subset of layers that work from previous literature (Skean et al, ICML 2025) that are effective for layer-wise experiments. Kindly clarify this please.
> > >
> > > Skean et al, ICML 2025 studies the intermediate layer representations and show that they (probably from layer between [25, 30]) can outperform the final layer on dowmstream tasks. However, what they study is more like the embedding task (e.g., MTEB, or representation distance/entropy on WikiText). On the other hand, our method let the intermediate layers to decode meaningful contents, such that the entire framework becomes a multi-level generator, instead of a purely representing model.

---

### Official Review · Reviewer_oURU · 2025-10-28

**Soundness:** 3
**Presentation:** 2
**Contribution:** 2
**Rating:** 4
**Confidence:** 4

**Summary:**

The paper introduces hierarchical decoding language models (HDLM), a post-hoc adaptation of decoder-only LLMs that replicates the final language head onto selected intermediate layers and trains those layers to decode intermediate responses.

**Strengths:**

The authors introduce the interesting setting of hierarchical decoding and present a simple, broadly applicable recipe that augments existing decoder-only LLMs without architectural redesign. The core mechanism of replicating the final language head onto intermediate layers and applying multi-layer decoding losses, turns a standard transformer into a model capable of producing multi-stage responses. This idea is well-motivated by prior observations that intermediate layers carry semantically rich representations and is implemented cleanly as a post-hoc hierarchical supervision method, requiring no retraining from scratch.

The evaluation spans a wide spectrum of benchmarks in hierarchical text classification (WoS, DBpedia), classification-guided dialog generation (ESConv, EmpatheticDialogues), and hierarchical reasoning/generation tasks (AQuA, CommonSenseQA, ToM). Across these diverse settings, HdLM consistently improves over supervised fine-tuning (SFT) and other baselines.

The theoretical FLOPs analysis is carefully derived and linked to empirical timing measurements. The paper explicitly computes training and inference complexity and demonstrates measurable compute savings that scale with the intermediate-layer index k and output length, aligning well with theoretical expectations.

**Weaknesses:**

The novelty is somewhat overstated. Replicating language heads onto intermediate layers for auxiliary supervision is conceptually close to existing multi-layer or hierarchical decoding methods, but no direct comparison is provided. Without including baselines like Hierarchical Skip Decoding (https://arxiv.org/abs/2404.16710) or intermediate-layer probing (https://arxiv.org/abs/2407.10795) variants, it’s hard to assess the relevance of the proposed method that requires task-specific tuning.

Further, the paper doesn’t clearly explain early on how intermediate-layer decoding aligns with the token-level autoregressive process. For example, Figure 1 is more confusing than helpful. One has to dig deep into the paper and learn how to parse complicated notation in order to understand what is going on.

The “proof” in Section 3.2 on convergence feels disconnected from the rest of the paper. It gestures toward theoretical depth but doesn’t establish any practically meaningful guarantee or insight about HdLM. The argument mostly reuses an existing theorem and reformulates it with different notation, without connecting it back to model behavior or experiments. A more focused empirical or ablation-based discussion of convergence behavior would be far more valuable.

Also, removing Section 3.2 would be a good option to create space for the related work section, which is far too short and reads more like a placeholder than a serious engagement with prior literature. Given the number of existing studies on intermediate-layer decoding, hierarchical supervision, and multi-level reasoning, this omission stands out. The paper would benefit from a more comprehensive positioning — not only citing prior work but actually clarifying where HdLM diverges conceptually and empirically. As it stands, this section undersells the novelty and gives the impression that the authors have not fully mapped the landscape.

The provided implementation and reproducibility details, while commendable, appear incomplete. The public repo’s folder structure doesn’t match the README, and the src directory required by the training script is missing. This undermines the otherwise strong claim of reproducibility. Overall, while the approach is well-motivated and shows consistent improvements, the paper needs clearer baselines, a more transparent decoding description, and complete reproducibility materials to reach the standard of a strong ICLR paper.

**Questions:**

How does the intermediate decoding process intuitively map to the partial responses in hierarchical decoding? A clear, high-level explanation in words (without equations) would help readers understand how the mechanism relates to standard autoregressive decoding.

Including baselines such as LayerSkip (https://arxiv.org/abs/2404.16710) would significantly strengthen the paper and directly improve my evaluation of its contribution. These comparisons are critical to demonstrate that HdLM provides a distinct advantage over existing adaptive or hierarchical decoding methods.

The related work section should be expanded substantially. Works on layer skipping, adaptive compute for decoding https://arxiv.org/abs/2312.16392, and mechanistic interpretability studies on intermediate decoding (https://arxiv.org/abs/2401.06102) are directly relevant. Addressing these connections would materially improve the paper and my rating.

Further fixing the code submission and addressing my raised weaknesses would improve my impression about the work.

---

> ### Author Response · Authors · 2025-11-25
>
> We sincerely appreciate your insightful feedback. Below are our respondes:
>
> > How does the intermediate decoding process intuitively map to the partial responses in hierarchical decoding? A clear, high-level explanation in words (without equations) would help readers understand how the mechanism relates to standard autoregressive decoding.
>
> We acknowledge that the methodology was not sufficiently clarified in the main text. Figure 1 was initially aimed to introduce the core paradigm of HdLM (i.e., leveraging distinct layers for hierarchical content processing), while Figure 3 details the implementation mechanism and training methods.
>
> Specifically, HdLM maintains full connectivity (self-attention and MLP) across all layers but introduces an additional patch-wise causal mask alongside the standard causal mask. The hidden state processes the full sequence [L_prev + L_d], though **only the L_prev segment contributes to loss computation and propagates to subsequent layers** — visualized in Figure 3 using cross-hatched grids. We regret that this critical design element was not adequately explained in the current manuscript and will revise both Figure 1 and Figure 3 to ensure comprehensive methodological clarity.
>
> > Including baselines such as LayerSkip (https://arxiv.org/abs/2404.16710) would significantly strengthen the paper and directly improve my evaluation of its contribution. These comparisons are critical to demonstrate that HdLM provides a distinct advantage over existing adaptive or hierarchical decoding methods.
>
> Thanks for pointing out this issue. To further strengthen our experimental results, we have added two recent baselines, either adaptive (LayerSkip [1]) or hierarchical decoding (SL-D [2]), with results on ESConv (as a case of CgG):
>
> | **Method**   | **ACC**         | **MaF1**     | **bias** | **B-2**  | **R-L** |
> | ------------------------- | ------------------------------- | ----------------------- |----------------------- | ----------------------- | ----------------------- |
> | LayerSkip [1]    |  32.43   |  21.29   |  1.28  |   6.97  |  16.59   |
> | SL-D [2]   |   28.00  |  23.70   |  0.41  |   5.88  |   15.30  |
> | **HdLM**    | 33.41    | 21.65    | 0.89   | 7.54   | 18.13   |
>
> and on HTG:
>
> | **Method**   | **AQuA** (ID)         | **CommonSenseQA** (OOD)     |
> | ------------------------- | ------------------------------- | ----------------------- |
> | LayerSkip [1]    |  38.5   |  71.3   |
> | SL-D [2]   |   38.2  |  67.5   |
> | **HdLM**    | 25.2    | 72.9    |
>
> Results show that HdLM can outerperform them on CgG tasks, and also on the OOD test. These results show that HdLM can perform well compared to these existing methods, especially on classification-related tasks.
>
> > The related work section should be expanded substantially. Works on layer skipping, adaptive compute for decoding https://arxiv.org/abs/2312.16392, and mechanistic interpretability studies on intermediate decoding (https://arxiv.org/abs/2401.06102) are directly relevant. Addressing these connections would materially improve the paper and my rating.
>
> We sincerely appreciate the reviewer's insightful suggestions. In the revised manuscript, we have thoroughly restructured the Related Work section to more clearly articulate the novelty and significance of our contribution.
>
> > The “proof” in Section 3.2 on convergence feels disconnected from the rest of the paper.
>
> We sincerely appreciate the reviewer's feedback. To optimize space for additional experimental results, we have relocated this section to the appendix in the revised manuscript.
>
> > The public repo’s folder structure doesn’t match the README, and the src directory required by the training script is missing. This undermines the otherwise strong claim of reproducibility.
>
> We have restructured the codebase to ensure full alignment with the README instructions, and the new repository URL (https://anonymous.4open.science/r/HdLM-2025) has been updated in the revised manuscript.
>
> ### References:
>
> [1] LayerSkip: Enabling Early Exit Inference and Self-Speculative Decoding, *ACL 2024*
>
> [2] Multilingual Contrastive Decoding via Language-Agnostic Layers Skipping, *ENMLP 2024*

---

> > ### Comment · Reviewer_oURU · 2025-11-27
> > **Request for clarifications**
> >
> > Thank you for the response and additional results.
> >
> > 1. Could you please fix the CgG table, it does not render correctly. Also are these performance differences meaningful? Is for AQuA smaller better? How large are these datasets.
> >
> > 2. I checked the manuscript and it has not been updated. Are you planning to update only in case of acceptance? I would like to see a preview of the updates.
> >
> > 3. My main remaining concern is the novelty and relevance of this work. How is this work different from the other intermediate decoding and layer skipping techniques? Also could you please explain why this setting of hierarchical language modeling is relevant and what does it add to simple autoregressive models that e.g. are cleverly prompted?

---

> > > ### Author Response · Authors · 2025-12-02
> > >
> > > 1. Sorry about that. I have fixed the CgG table.
> > > For both AQuA and CommonSenseQA, the larger is better. That means our method has a lower accuracy than LayerSkip and SL-D on AQuA (in-domain, 100k samples), but has a better performance on CommonSenseQA (out-of-domain, 10k samples).
> > > 2. We have updated the repo with new anonymou link (https://anonymous.4open.science/r/HdLM-2025), and the original link in the submission version of paper has been forfeited. We have changed the code structure, and make it consistent with the command in README (e.g. train/LLaMA-Factory/training_scripts/wos_train.sh).
> > > 3. We have updated the Related Work Section and highlight the differrence between our method and previous layerskipping methods (LayerSkip, SL-D, Medusa).
> > >
> > > For example, LayerSkip enables the LLM to decode self-speculatively and inference with early exit; SL-D implements the layer skipping with layer contrastive decoding; Medusa implements multiple decoding heads in parallel. On the other hand, our generates different level of abstractions, in contrast with speculative decoding methods; furthermore, HdLM is built on pretrained LLMs, with stronger capabilites and more levels of decoding hierarchies, compared to HRM.
> > >
> > > HdLM has also been proved to outperform than prompt-based method, which are CoT, Self-Refine, and FSM, in the original version of paper. Table 1, 2, 3 verify the superiority of HdLM on our HTC, CgG and HTG experiments.

---

### Official Review · Reviewer_3dxA · 2025-10-29

**Soundness:** 3
**Presentation:** 3
**Contribution:** 3
**Rating:** 4
**Confidence:** 4

**Summary:**

This paper introduces the Hierarchical decoding Language model (HdLM), a new architecture designed to embed hierarchical reasoning capabilities directly into a language model. Motivated by human cognition—which moves from coarse-grained strategy to fine-grained action—the method adapts a pre-trained decoder (Llama3-8B) by copying its final language head to selected intermediate layers. These new heads are then fine-tuned to decode specific, intermediate steps of a complex task in a streaming, layer-by-layer fashion.
The authors evaluate HdLM on three categories of hierarchical tasks: Hierarchical Text Classification (HTC), Classification-guided Generation (CgG), and Hierarchical Text Generation (HTG). The empirical results are strong, showing that HdLM achieves state-of-the-art performance on multiple benchmarks and, notably, outperforms much larger models like GPT-4 on specific reasoning tasks. The paper also provides a theoretical analysis of the method's computational efficiency gains during training and inference.

**Strengths:**

- The paper's primary contribution is moving beyond prompting (like CoT) or latent-space manipulation. It introduces a powerful structural inductive bias for reasoning by physically modifying the model's architecture. Aligning computational depth  with conceptual abstraction  is a fundamental and intuitive idea that provides a more native solution for multi-step tasks.
- The experimental results are highly compelling. The fact that an 8B-parameter HdLM can outperform GPT-4 on complex reasoning (ToM tests) is a significant finding. It provides a strong argument that intelligent architectural design can be more effective and efficient than brute-force scaling, a highly relevant conclusion for the field.

**Weaknesses:**

- The method's design fundamentally requires the number of hierarchical steps (D) to be strictly less than the total model layers (K). This hard constraint (D < K), makes the model incapable of handling tasks with long, variable, or arbitrarily deep reasoning chains (e.g., D > K). This is a scenario where traditional CoT methods, which decode all steps at the final layer, remain far more flexible.
- The model's performance hinges critically on two sets of hyperparameters: the intermediate layer indices (k_d) and their corresponding loss weights (f_d). The paper provides no systematic, principled method for selecting these. The sensitivity analysis is limited to D=2, and the choice for D=3 (k=[20, 30]) appears entirely empirical. This reliance on expensive, task-specific manual tuning severely limits the method's scalability and generalization to new, unseen tasks.
- The HdLM architecture enforces a strict, unidirectional (shallow-to-deep) information flow.  This makes the model highly fragile: any error generated at an early layer (k_1) is irreversibly passed on and likely amplified at subsequent layers. This is a critical flaw that is masked by the paper's experimental design. By evaluating only on shallow tasks (D=2 or D=3), the impact of this error accumulation is not apparent. It is highly probable that on deeper, long-chain reasoning tasks (e.g., D=10), HdLM's performance would collapse due to this cumulative error, potentially performing much worse than a more robust CoT baseline that theoretically allows for self-correction.
- The paper is critically ambiguous about its core mechanism, hindering reproducibility. It fails to clearly explain how information is passed between layers. It's unclear how a newly decoded discrete text r_d is integrated with a continuous hidden state e_d for processing by subsequent layers. The (presumed) mechanism—that the sequence length (L) of the hidden state grows ([L_prev + L_d, E]) while the hidden dimension (E) remains constant—is never explicitly stated.

**Questions:**

- For tasks with D > 2, is there any systematic or principled method to select the layer indices (k_1...k_{D-1}) and loss weights (f_1...f_{D-1})? Or is this purely a matter of empirical, task-specific tuning?
-  Please explicitly confirm the information flow mechanism during inference. Is it correct to assume that the hidden states e'_d (shape [L_d, E]) corresponding to the decoded tokens r_d are concatenated with the input hidden states e_d (shape [L_prev, E]) to form a new, larger tensor of shape [L_prev + L_d, E], which is then fed as input to the next block of layers M_{k_d:k_{d+1}}?
- How do the authors justify the D < K hard limit versus the flexibility of CoT?
- Given that experiments were run only on shallow (D=2, 3) tasks, what evidence or argument do the authors have that HdLM would not suffer from catastrophic performance collapse on deeper tasks (e.g., D=10) due to the error propagation inherent in its unidirectional architecture?
- Given the high risk of error propagation, have the authors considered any mechanisms to improve robustness, such as a feedback loop that allows a deeper layer's state to correct or refine the output of a shallower layer?

---

> ### Author Response · Authors · 2025-11-25
>
> > is there any systematic or principled method to select the layer indices?
>
> As studied by [1], middle layers encode valuable representations for downstream tasks, and the 32-layer LLaMA-8B model can be partitioned into three segment:
> -the region of **shared layers (1-9)**
> -the region of **transition layers (10-15)**
> -the region of **refinement layers (16-32)**
> therefore, the optimal layer selection should fall within the refinement region (16 ≤ k ≤ 32).
>
> Futhermore, [2] observes that LLM first undergoes a *`compression valley`* where entropy decreases until k = 25; [3] identifies k = 25 as the intersection point between *concept generation* and *token generation* phases. These findings  allow us to finally determine the optimal k with reduced costs: **25 ≤ k ≤ 30**.
>
> Our empirical findings in Figure 4 (Section 4.4) reveals that: HTC (WoS) accuracy remains robust across D within this region; while CgG exhibits a clear trade-off requiring more layers for generation (e.g., k = 28). These observations lead us to conclude that **classification tasks require fewer layers than generation tasks** (mentioned in Section 4.4). As a result, **k of CgG and HTG should be sligntly larger than k of HTC**, and the formal settings **k=25 for WoS and k=28 for ESConv** aligns precisely with the theoretical guidance from [1]–[3].
>
> >  Is it correct to assume that the hidden states e'd (shape [L_d, E]) corresponding to the decoded tokens r_d are concatenated with the input hidden states e_d (shape [L_prev, E]) to form a new, larger tensor of shape [L_prev + L_d, E], which is then fed as input to the next block of layers M{k_d:k_{d+1}}?
>
> Yes, you are right. The inference of HdLM have two propagation mechanism simultanously: for some layer with decoding functionality, it generates L_d latents, then i) decodes to L_d tokens by its heads; and ii) concatenated with L_prev latent, then pass to the next block of layers M{k_d:k_{d+1}}. We apologize that this critical design was not adequately explained in the main text, and we will thoroughly revise Section 2 to clarify this mechanism with precise technical descriptions.
>
> > How do the authors justify the D < K hard limit versus the flexibility of CoT?
>
> HdLM addresses two hierarchical mechanisms:
> - (i) coarse-to-fine abstractions (e.g., high-level plans/slow thinking $\rightarrow$ fine-grained actions/fast thinking)
> - (ii) suquential determinations (e.g. different reasoning steps)
> Our experimental focus primarily the mechanism (i), which typically involves 2–3 abstraction levels. We note that practical limitations prevent conceptual abstractions from exceeding 32 (D > 32 could be infeasible for human's abstraction capabilities).
>
> For mechanism (ii), we acknowledge that HdLM is not designed for extended reasoning chains—though this falls outside the primary scope of our work. We deeply regret the misleading example in Figure 1 and will replace it with examples explicitly demonstrating mechanism (i) in the revised manuscript.
>
> > what evidence or argument do the authors have that HdLM would not suffer from catastrophic performance collapse on deeper tasks (e.g., D=10)
>
> This challenge is not unique to HdLM — it similarly manifests in traditional LLM reasoning chains. For instance, as noted in [4], "initial errors often propagate and undermine the reliability of final conclusions, while current LLM-based error detection methods struggle to identify such propagated errors due to corrupted downstream judgments." Similarly, [5] states that "existing uncertainty estimation techniques primarily focus on final-step outputs, failing to account for cumulative uncertainty across multi-step reasoning." Thus, HdLM and conventional CoT reasoning share the same fundamental limitation regarding the inductive bias of early-decoded tokens. This issue represents a broader challenge in LLM reasoning that warrants dedicated future work for resolution.
>
> > have the authors considered any mechanisms to improve robustness, such as a feedback loop that allows a deeper layer's state to correct or refine the output of a shallower layer?
>
> Specifically, we can implement a layer-wise switcher leveraging speculative decoding at each layer (integrating the framework of LayerSkip): if the current layer's decoding is correct, we retain the intermediate decoding solution; otherwise, we bypass intermediate decoding and utilize the intermediate latent solely to guide subsequent layer generation. We will highlight these potential extensions in the revised manuscript.
>
>
> ### References:
>
> [1] Layer by Layer: Uncovering Where Multi-Task Learning Happens in Instruction-Tuned Large Language Models, *EMNLP 2024*
>
> [2] Layer by Layer: Uncovering Hidden Representations in Language Models, *ICML 2025*
>
> [3] Multilingual Contrastive Decoding via Language-Agnostic Layers Skipping, *EMNLP 2024*
>
> [4] Probabilistic Soundness Guarantees in LLM Reasoning Chains, *EMNLP 2025*
>
> [5] Uncertainty Propagation on LLM Agent, *ACL 2025*

---

### Official Review · Reviewer_Q9Gc · 2025-11-01

**Soundness:** 2
**Presentation:** 3
**Contribution:** 3
**Rating:** 6
**Confidence:** 3

**Summary:**

This paper propose Hierarchical decoding Langauge Model (HdLM), which enables different intermediate transformer layers to decode text hierarchically. Pretrained model can be adapted to HdLM by copying the language heads of the last layer to different selected intermediate layers, and fine-tunining with different task inputs. The paper evaluates HdLM’s performance on three different tasks: hierarchical text classification (HTC), classification-guided generation (CgG), and hierarchical text generation (HTG). HdLM outperforms baselines on multiple datasets including DBpedia, AQuA, and CommonSenseQA in some metrics.

**Strengths:**

- The idea of enabling intermediate layers to decode is creative and well-motivated by human hierarchical reasoning capabilities.
- The paper experiments with multiple datasets, and achieve performance that can surpass GPT-4 + CoT / SimTom with a much smaller model in HTG tasks.
- The paper conduct ablation on the sensitivity in the intermediate decoding layers choices, scalability on model sizes, and computational time.

**Weaknesses:**

- The automatic evaluation metrics used in this paper (BLEU and Rouge-L) are more reliable when evaluating tasks with significant lexical overlap, but might not be appropriate to serve as a quality metric in this paper’s setting. Furthermore, the absolute values of BLEU-2 scores are very low (<6). Although the performance of the classification task is higher, I am not convinced that the improvement in empathetic dialogues is significant. I would love to see LLM-as-a-judge / human evaluation results.
- Although theoretically speaking, this framework can be adapted to depth D that < K, this also requires the users to know the value of D, and need to tune the selected layers accordingly. Based on figure 4,  the performance is sensitive of the intermediate layer choices. However, the paper provides no principled selection method, making HdLM significantly more difficult to apply than SFT in practice.

**Questions:**

- Quotes on line 37: try `` and ''
- Last line of page 1 doesn’t look like a finished sentence?
- In Table 1, SFT’s MaF1 is better than HdLM’s MaF1
- In Table 2, SFT+FSM on B-2 is actually the best, rather than +HdLM
- Why isn’t SFT+Self-Refine tested in table 2? Direct+Self-refine is the best performing on for direct settings in terms of ACC and MaF1.
- How does the training converge?

---

> ### Author Response · Authors · 2025-11-25
>
> > I am not convinced that the improvement in empathetic dialogues is significant. I would love to see LLM-as-a-judge / human evaluation results.
>
> We agree that human evaluation and LLM-as-a-judge are reliable tools to evaluate the dialogue responses. To further verify the performance of HdLM, we conduct new evaluations on ESConv, with human evaluation results shown below:
>
> | **Method**   | **Fluecy**         | **Emotion** | **Effectiveness** | **Satisfaction** |
> | ------------------------- | ------------------------------- | ---------------- | ----------------------- |----------------------- |
> | Direct    | 2.95    | 3.00           | 2.40        |  2.60   |
> | + Self-Refine  | 3.10  | 3.15   | 2.70        |  2.80    |
> | + CoT  | 3.08    | 3.08           | 2.67         |   2.83    |
> | + FSM   |      3.30       |   3.35    |    2.90   |      2.93    |
> | ------------------------- | ------------------------------- | ---------------- | ----------------------- |----------------------- |
> | SFT    | 3.15   | 3.40           | 2.70           | 2.90    |
> | + CoT  | 3.67        |  3.61       |  3.67         |  3.45    |
> | + FSM  | 3.80       | 3.55           | 3.70          |  3.65    |
> | **HdLM (Ours)**    |      3.84       | 3.67    | 3.97| 3.86    |
>
> with 4 human annotators; annotation principles can be refer to [1].
>
> And also pairwise results evaluated by GPT4o, with the evaluation prompt from [2]:
>
> | **Method**   | **win**         | **tie** | **lose** |
> | ------------------------- | ------------------------------- | ----------------------- |----------------------- |
> | FSM VS SFT    | 59.5    | 11.1     | 29.4   |
> | **HdLM** VS SFT    | 61.5    | 11.7    | 26.8   |
>
> Both results show that HdLM can perform better than finetuned baselines.
>
> > the paper provides no principled selection method
>
> The depth (D) is predetermined by the tasks (2 or 3 in this paper); while we need to determine the decoding intermediate layer (k). As studied by [3], middle layers encode valuable representations for downstream tasks, and LLaMA-8B (32-layer) can be partitioned into three segments:
> -region of **shared layers (1-9)**
> -region of **transition layers (10-15)**
> -region of **refinement layers (16-32)**
> therefore, the optimal k should fall within the refinement region (16 ≤ k ≤ 32).
>
> Futhermore, [4] observes that LLM first undergoes a *`compression valley`* where entropy decreases until k = 25, followed by entropy increase; [5] identifies k = 25 as the intersection point between the *concept generation* and *token generation* phases. These findings collectively justify narrowing the optimal k range to around **25 ≤ k ≤ 30**.
>
> The above narrowed region allows us to finally determine the optimal k with reduced costs. Our empirical findings in Figure 4 (Section 4.4) reveals that: HTC (WoS) accuracy remains robust across k within this region; while CgG exhibits a clear trade-off requiring more layers for generation (e.g., k = 28). These observations lead us to conclude that **classification tasks require fewer layers than generation tasks** (mentioned in Section 4.4). As a result, **k of CgG and HTG should be sligntly larger than k of HTC**, and the formal settings **k=25 for WoS and k=28 for ESConv** aligns precisely with the theoretical guidance from [1]–[3]. We will emphasize this alignment in the revised manuscript.
>
> > Why isn’t SFT+Self-Refine tested in table 2?
>
> We observe that SFT+Self-Refine underperforms relative to the SFT baseline. We suppose that Self-Refine, as a multi-hop inference paradigm, inherently requires additional multi-turn samples when integrated with fine-tuning. In contrast, all other baselines—including our HdLM—are trained exclusively on the ESConv training set. Expanding the training set for SFT+Self-Refine would introduce an unfair advantage and complicate the comparison. In the revised manuscript, we will clarify these points in Section 4.3.
>
> > How does the training converge?
>
> Due to page constraints, we present the loss analysis in Appendix D.2 (Fig. 5 and Fig. 6), demonstrating that HdLM training converges after approximately 300 steps, with stable intermediate losses (<$L_1$>) decreasing as the intermediate layer index (k) increases. These findings corroborate the theoretical claims in Corollary 2 and indicate that higher layer depths should be reserved for more challenging tasks, aligning with our observations in Section 4.3.
>
> ### References:
>
> [1] Can Large Language Models be Good Emotional Supporter? Mitigating Preference Bias on Emotional Support Conversation, *ACL 2024*
>
> [2] Steering Conversational Large Language Models for Long Emotional Support Conversations, *SICon 2025*
>
> [3] Layer by Layer: Uncovering Where Multi-Task Learning Happens in Instruction-Tuned Large Language Models, *EMNLP 2024*
>
> [4] Layer by Layer: Uncovering Hidden Representations in Language Models, *ICML 2025*
>
> [5] Multilingual Contrastive Decoding via Language-Agnostic Layers Skipping, *EMNLP 2024*

---

### Meta-Review · Area_Chair_hGpP · 2026-01-13

**Summary:**

This paper proposes a hierarchical decoding architecture for language models where intermediate transformer layers decode text at different levels of abstraction. The method copies language heads from the final layer to selected intermediate layers and fine-tunes them. The authors evaluate their approach on three tasks (hierarchical text classification, classification-guided generation, and hierarchical text generation). The paper reports improvements over supervised fine-tuning baselines across multiple benchmarks and provides theoretical analysis of computational savings.

**Reviews and Rebuttal.** This paper received four reviews with lukewarm initial scores (6, 4, 4, 4). Reviewers found the idea interesting and noted the breadth of experiments and the compute motivation. The main concerns focused on (i) heavy reliance on manual design choices, especially the selection of intermediate layer indices and loss weights, and (ii) lack of clarity about the mechanism by which intermediate decoded tokens and representations are passed forward and used by later layers. The rebuttal added some additional evaluation and clarifications, but it did not substantially change the assessment that the method depends on task-specific tuning and that the core mechanism needs a cleaner, more explicit specification.

**Recommendation.** Having read the reviews, rebuttal and paper – I am unfortunately recommend rejection at this stage. Overall, I find that the the paper presents an appealing architectural concept, but suffers from a number of concerns that undermine soundness and significance. As it stands, the evidence suggests that performance depends critically on manual choices for layer selection and loss weighting (and could use guidance for how to choose these in a principled way across tasks). Moreover, the mechanism for how intermediate decoding interacts with subsequent computation remains unclear. In some sense both of these choices could be mitigated by some work by a camera-ready, but they are the kinds of changes that merit another round of reviews.

**Reviewer Concerns:**

Outstanding:

- Mechanism (how intermediate decoding interacts with subsequent computation) [3dxA, oURU]
- Design Choices (e.g., layer selection, loss weights; task-specific tuning) [3dxA, oURU, 7VJP]
- Positioning (e.g., layer-skipping, adaptive decoding literature) [7VJP, oURU]

Resolved or Ignored:

- Baselines / Evaluation (LayerSkip/SL-D added) [oURU, Q9Gc]
- Reproducibility (repository issues fixed) [oURU]

**Reviewer Scores:**

I expect scores would have shifted from (6, 4, 4, 4) -> (6, 5, 4, 5). oURU and 7VJP would likely increase by one point given the added baselines and clarifications. 3dxA would maintain at 4 given unresolved concerns about mechanism and design choices. Q9Gc would maintain at 6.

---

### Decision · Program_Chairs · 2026-01-26

Reject